# NICE: NoIse-modulated Consistency rEgularization for Data-Efficient GANs

**Yao Ni**[†], **Piotr Koniusz**[*,§,†]
[†]The Australian National University    [§]Data61♥CSIRO
[†]firstname.lastname@anu.edu.au

## Abstract

Generative Adversarial Networks (GANs) are powerful tools for image synthesis. However, they require access to vast amounts of training data, which is often costly and prohibitive. Limited data affects GANs, leading to discriminator overfitting and training instability. In this paper, we present a novel approach called NoIse-modulated Consistency rEgularization (NICE) to overcome these challenges. To this end, we introduce an adaptive multiplicative noise into the discriminator to modulate its latent features. We demonstrate the effectiveness of such a modulation in preventing discriminator overfitting by adaptively reducing the Rademacher complexity of the discriminator. However, this modulation leads to an unintended consequence of increased gradient norm, which can undermine the stability of GAN training. To mitigate this undesirable effect, we impose a constraint on the discriminator, ensuring its consistency for the same inputs under different noise modulations. The constraint effectively penalizes the first and second-order gradients of latent features, enhancing GAN stability. Experimental evidence aligns with our theoretical analysis, demonstrating the reduction of generalization error and gradient penalization of NICE. This substantiates the efficacy of NICE in reducing discriminator overfitting and improving stability of GAN training. NICE achieves state-of-the-art results on CIFAR-10, CIFAR-100, ImageNet and FFHQ datasets when trained with limited data, as well as in low-shot generation tasks.

## 1   Introduction

The remarkable advancements and breakthroughs in deep learning [31] can be largely attributed to the extensive utilization of vast amounts of training data. This abundance of data has driven the progress across various domains of deep learning. Among notable recent advancements are Generative Adversarial Networks (GANs) [13], popular in industry and academia. GANs have proven their high-quality image generation abilities and achieved high generation speeds [47], establishing them as a versatile tool for a wide range of applications, such as text-to-image generation [20, 47, 55], destylization [48, 49, 50], image-to-image translation [27, 44], and 3D generation [52, 53, 65].

Despite the impressive capabilities of state-of-the-art GANs in generating diverse and high-quality images [7, 25], their effectiveness heavily relies on large volumes of training data. The acquisition of such large datasets helps GANs attain the desired adversarial equilibrium. However, under limited training data regime, GANs encounter challenges associated with discriminator overfitting and unstable training [22, 68, 57, 11, 17].

To address the aforementioned challenges, recent studies have approached the problem from three perspectives. The first perspective involves the utilization of extensive differentiable data augmentation techniques, aimed at expanding the distribution of the available data [22, 68, 17, 32, 63, 64, 69]. The second perspective leverages knowledge gained from large-scale models trained on large datasets [9, 70, 29]. While these approaches are effective, they also carry inherent risks, such as the leakage of augmentation clues [64, 22, 40] or pre-trained knowledge [30].

---

[*]The corresponding author.   Code: `https://github.com/MaxwellYaoNi/NICE`

37th Conference on Neural Information Processing Systems (NeurIPS 2023).

The third perspective focuses on the regularization of the discriminator, aiming to weaken its learning ability [11, 19] or increase the overlap between real and fake support sets [57]. In line with this perspective, our paper introduces a novel approach to improve GAN generalization. We propose to modulate the hidden features of discriminator via adaptive multiplicative noise, which strikes a balance in maintaining a certain level of discrimination ability while also regularizing the Rademacher complexity [6] of the discriminator. This reduction in Rademacher complexity, which quantifies the capacity of model to fit random variables, narrows the generation gap between the training and unseen data, resulting in enhanced GAN generalization [4, 56, 18].

Nevertheless, training the discriminator with adaptive noise unintentionally amplifies second-order gradient derivative of latent features corresponding to real images. This elevated gradient leads to abrupt gradient changes near the real sample points, potentially causing instability in the feedback to the generator. This issue aligns with the findings in works [26, 36, 56, 11], which emphasize the importance of penalizing gradients for both real and fake samples to promote convergence and stability of GAN training. To address these challenges, we propose a constraint on the discriminator that ensures consistency for the same inputs under different noise modulations. While our idea is simple, our theoretical analysis reveals that this constraint effectively penalizes the first and second-order gradients of the latent features. Consequently, the gradient provided to the generator becomes more stable, resulting in improved training stability and generalization.

Our comprehensive experiments confirm the effectiveness of NICE in penalizing gradients and reducing the generalization gap. Despite the simple design, NICE significantly improves the training stability and generalization of GANs, outperforming alternative approaches in preventing discriminator overfitting. NICE achieves superior results on challenging limited data benchmarks, including CIFAR-10/100, ImageNet, FFHQ, and low-shot image generation tasks.

Our contributions can be summarized as follows:

i. We limit discriminator overfitting by using the adaptive multiplicative noise to modulate the latent features of the discriminator, resulting in enhanced generalization of GANs.

ii. We introduce NICE, a technique that enforces the discriminator to be consistent with the same inputs under different noise modulations, implicitly penalizing the first and second-order gradients of the latent features during GAN training, promoting the training stability.

iii. We show that NICE, both in theory and practice, effectively prevents discriminator overfitting and achieves superior performance in image generation under limited data setting.

## 2 Related Work

**Improving GANs.** Generative Adversarial Networks [13] are powerful generative models that excel in image generation [21, 24, 25], text-to-image generation [20, 47, 55], image-to-image translation [27, 44], and 3D generation [52, 65, 53]. However, GANs commonly encounter challenges such as training instability [26], mode collapse [45], and discriminator overfitting [22]. Researchers have investigated different GAN loss functions [2, 42, 71], architectures [21, 24, 25, 23], and regularization strategies [38, 33, 14]. The $f$-GAN [42] generalizes GANs to $f$-divergences. WGAN [2] adopts the Earth-Mover distance. OmniGAN [71] extends conditional GANs to a multi-label softmax loss. StyleGANV1-3 [24, 25, 23] enhances the architecture of generator. Approaches [14, 26, 36, 56, 11] propose explicit penalty of gradient of discriminator. SNGAN [38] enforces a Lipschitz constraint on the discriminator, and approach [33] regularizes the spectral norm of the discriminator. In this study, we propose a novel regularization strategy applicable to diverse GAN loss functions and architectures, specifically tailored for training GAN models with limited data.

**Image generation with limited data.** Collecting data is both laborious and expensive. Consequently, generating images in limited data settings presents significant challenges, primarily due to discriminator overfitting. Previous approaches have tackled this issue through data augmentation techniques [22, 68, 17, 32, 63] and leveraging knowledge transfer from large-scale pre-trained models [9, 29, 70]. Differentiable data augmentation methods [22, 68, 17] "extend" the data distribution, while methods [63, 32] explore contrastive learning within GANs. Approaches [9, 29, 70] employ pre-trained models to guide training of discriminator. However, both GAN types suffer issues, including leakage of augmentation clues [64, 22] or pre-trained knowledge [30]. An alternative approach involves regularizing the discriminator. LeCamGAN [57] suggests reducing the output gap of discriminator between real and fake distributions. LCSAGAN [40] employs manifold techniques to project discriminator features onto manifold and decrease the capacity of discriminator. APA [19]

uses fake images as real images to increase overlap between the real and fake distributions. DigGAN [11] minimizes the gradient norm gap between real and fake images. Inspired by such an approach, we propose to use an adaptive noise to modulate the latent features of discriminator to reduce the generalization error of discriminator as a means of regularization.

**Consistency regularization.** CTGAN [61] introduces the enforcement of a Lipschitz constraint by ensuring the consistency of the response of discriminator to real images. CAGAN [41] and GenCo [8] enforce consistency among multiple discriminators. R-Drop [62] applies consistency regularization to transformers [58] equipped with dropout for natural language processing tasks. Augmentation-based consistency regularization GANs [64, 69] enforce consistency by considering different augmented views of the same image, although they may inadvertently leak augmentation clues to the generator [22]. Despite the effectiveness of consistency regularization demonstrated by these previous works, their success is primarily empirical, lacking theoretical analysis. In our study, we go beyond prior works by providing a theoretical analysis of consistency regularization in GANs. By delving deeper into the underlying principles, we develop a more comprehensive understanding of the mechanism behind its effectiveness.

**Generalization of GANs.** Arora *et al.* [4, 56, 67, 18] have contributed to the understanding and improvement of generalization of GANs by showing the importance of achieving adversarial equilibrium, reducing discriminator discrepancy, penalizing gradients, and bounding the generalization error. Our motivation aligns with these works in the pursuit of enhancing generalization capabilities. However, our approach implicitly and adaptively reduces the generalization gap while penalizing gradients of latent features of discriminator. Such a setting provides an efficient and effective means of preventing discriminator overfitting and improving generalization.

## 3 Method

To boost the generalization of GAN, we start by analyzing their generalization error, bounding it with the Rademacher complexity of discriminator, and linking it with the weight norm. By incorporating multiplicative noise, we demonstrate its regularization benefits on the weight norm. However, such a strategy induces large gradient norm destabilizing training of GAN. To this end, we introduce NICE to penalize the gradients of discriminator. Finally, we provide NICE and showcase it use in GANs.

### 3.1 Generalization error of GANs and complexity of neural network

The primary goal of GAN is to minimize the integral probability metric [39], assuming access to infinite real and fake data during optimization, *i.e.*, the infinite real and generated distributions $(\mu, \nu)$ as discussed in [67]. In practice, we often have limited access to a finite dataset $\hat{\mu}_n$ of size $n$. Consequently, our optimization is restricted to the empirical loss:

$$\inf_{\nu \in \mathcal{G}} \left\{ d_{\mathcal{H}}(\hat{\mu}_n, \nu) := \sup_{h \in \mathcal{H}} \{ \mathbb{E}_{\boldsymbol{x} \sim \hat{\mu}_n}[h(\boldsymbol{x})] - \mathbb{E}_{\tilde{\boldsymbol{x}} \sim \nu}[h(\tilde{\boldsymbol{x}})] \} \right\}. \tag{1}$$

Function sets of discriminator and generator, $\mathcal{H}$ and $\mathcal{G}$, are typically parameterized in GAN as neural network classes $\mathcal{H}_{nn} = \{ h(\boldsymbol{x}; \boldsymbol{\theta}_d) : \boldsymbol{\theta}_d \in \boldsymbol{\Theta}_d \}$ and $\mathcal{G}_{nn} = \{ g(\boldsymbol{z}; \boldsymbol{\theta}_g) : \boldsymbol{\theta}_g \in \boldsymbol{\Theta}_g \}$ where $\boldsymbol{z} \sim p_z$ serves as the random noise input to the generator. The associated term $d_{\mathcal{H}_{nn}}$ is referred to as the neural network distance [4]. The discriminator network $D := \varphi \circ f$ consists of a real/fake prediction head $\varphi$ and a feature extractor $f$. As the loss function $\phi(\cdot)$ varies across tasks, architectures or choice of divergence type, we compose it with $D$ [2, 67, 4] to simplify the analysis and notation, *i.e.*, $h(\cdot) := \phi(D(\cdot))$. Thus, the alternative optimization of discriminator and generator becomes:

$$\begin{cases} L_D = \min_{\boldsymbol{\theta}_d} \mathbb{E}_{\tilde{\boldsymbol{x}} \sim \nu_n}[h(\tilde{\boldsymbol{x}}; \boldsymbol{\theta}_d)] - \mathbb{E}_{\boldsymbol{x} \sim \hat{\mu}_n}[h(\boldsymbol{x}; \boldsymbol{\theta}_d)], \\ L_G = \min_{\boldsymbol{\theta}_g} -\mathbb{E}_{\boldsymbol{z} \sim p_z}[h(g(\boldsymbol{z}; \boldsymbol{\theta}_g))], \end{cases} \tag{2}$$

where we assume $\nu_n$ minimizes $d_{\mathcal{H}}(\hat{\mu}_n, \nu)$ up to precision $\epsilon \geq 0$, meaning that $d_{\mathcal{H}}(\hat{\mu}_n, \nu_n) \leq \inf_{\nu \in \mathcal{G}} d_{\mathcal{H}}(\hat{\mu}_n, \nu) + \epsilon$. As we are interested in how close the generator distribution $\nu_n$ is to the unknown infinite distribution $\mu$, we refer to the lemma [67] on the generalization error of GAN:

**Lemma 1** *(Theorem 3.1 of [67]) Assume that the discriminator set $\mathcal{H}$ is even ($h \in \mathcal{H}$ implies $-h \in \mathcal{H}$) and all discriminators are bounded by $\|h\|_\infty \leq \Delta$. Let $\hat{\mu}_n$ be an empirical measure of an i.i.d. sample of size $n$ drawn from $\mu$. Assume $d_{\mathcal{H}}(\hat{\mu}_n, \nu_n) - \inf_{\nu \in \mathcal{G}} d_{\mathcal{H}}(\hat{\mu}_n, \nu) \leq \epsilon$. Then with probability at least $1 - \delta$, we have:*

$$d_{\mathcal{H}}(\mu, \nu_n) - \inf_{\nu \in \mathcal{G}} d_{\mathcal{H}}(\mu, \nu) \leq 2 \sup_{h \in \mathcal{H}} \left| \mathbb{E}_\mu[h] - \mathbb{E}_{\hat{\mu}_n}[h] \right| + \epsilon \leq 2 R_n^{(\mu)}(\mathcal{H}) + 2\Delta \sqrt{\frac{2 \log(1/\delta)}{n}} + \epsilon, \tag{3}$$

where the Rademacher complexity [6], $R_n^{(\mu)}(\mathcal{H}) := \mathbb{E}\big[\sup_{h\in\mathcal{H}} \frac{2}{n}\sum_i \tau^{(i)} h(\boldsymbol{x}^{(i)})\big]$, measures how well the function $h \in \mathcal{H}$ fits the Rademacher random variable $\tau^{(i)}$ with $prob(\tau^{(i)} = 1) = prob(\tau^{(i)} = -1) = \frac{1}{2}$ given samples $\boldsymbol{x}^{(i)} \sim \hat{\mu}_n$.

Lemma 1 provides a crucial insight that one can assess the generalization error of GAN by comparing the output discrepancy of discriminator between training data and unseen data, and such an error is influenced by the Rademacher complexity of the discriminator. To enhance the performance of generator while reducing the generalization error, we have two possibilities: 1) increase the quantity $n$ of real data, which provides a stronger foundation for training a better generator; 2) reduce the Rademacher complexity of the discriminator. However, this reduction must be carefully controlled, as an overly simplified discriminator may struggle to effectively distinguish real and fake data.

To manage the Rademacher complexity of the discriminator, we leverage a theorem from approach [5] to establish an upper bound on the Rademacher complexity of the neural network.

**Lemma 2** *(Eq. 1.2 in [5], Theorem 5.20 in [35])* *Consider a fully-connected neural network $v_{\boldsymbol{\theta}}(\boldsymbol{x}) = \boldsymbol{W}_t\sigma(\boldsymbol{W}_{t-1}\sigma(...\sigma(\boldsymbol{W}_1\boldsymbol{x})...))$ where $\boldsymbol{W}_i$ are linear weights at $i$-th layer and $\sigma$ is a 1-Lipschitz activation function. Suppose that $\forall i \in \{1,...,n\}$, $\|\boldsymbol{x}^{(i)}\|_2 \leq q$. Let $\|\boldsymbol{W}_i\|_{lip}$ be the lipschitz constant of $\boldsymbol{W}_i$ and $\|\boldsymbol{W}_i^T\|_{2,1}$ be the sum of the $l_2$ norm of columns in $\boldsymbol{W}_i$. Let $\mathcal{V} = \{v_{\boldsymbol{\theta}} : \|\boldsymbol{W}_i\|_{lip} \leq k_i, \|\boldsymbol{W}_i^T\|_{2,1} \leq b_i\}$, we have the Rademacher complexity:*

$$R_n^{(\mu)}(\mathcal{V}) \leq \frac{q}{\sqrt{n}} \cdot \left(\prod_{i=1}^t k_i\right) \cdot \left(\sum_{i=1}^t \frac{b_i^{2/3}}{k_i^{2/3}}\right)^{3/2}. \tag{4}$$

Lemma 2 links the Rademacher complexity of a neural network to the Lipschitz constant and the (2,1)-norm of its weights, providing essential insights for controlling the complexity of discriminator.

### 3.2 Improving generalization by feature modulation with multiplicative noise

Taking into account Lemmas 1 and 2, one can effectively manage the generalization error of GAN by controlling the Rademacher complexity of the discriminator. Such a control requires managing the Lipschitz constant and the norm of the weights of discriminator. Typically, controlling the Lipschitz constant can be achieved by the spectral normalization [38] or the gradient penalty [14]. In contrast, we propose a novel approach: regularizing the norm of the weights by modulating the latent features of the discriminator with multiplicative noise.

We build on the analysis (Prop. 4 in Appendix of [3]) of Dropout regularization [54] by exploring the regularization effect of the Gaussian multiplicative noise within the context of deep regression.

**Theorem 1** *(Regularization by the Gaussian multiplicative noise in deep regression)* *Consider a regression task on a two-layer neural network. Let $\mathcal{X} \subseteq \mathbb{R}^{d_0}$ and $\mathcal{Y} \subseteq [-1,1]^{d_2}$ be the input and output spaces, and $\mathcal{D} = \mathcal{X} \times \mathcal{Y}$. Let $n$ examples $\{(\boldsymbol{x}^{(i)}, \boldsymbol{y}^{(i)})\}_{i=1}^n \sim \mathcal{D}^n$. Let $f_w : \mathcal{X} \to \mathcal{Y}$ be parameterized by $\{w : \boldsymbol{W}_1 \in \mathbb{R}^{d_1 \times d_0}, \boldsymbol{W}_2 \in \mathbb{R}^{d_2 \times d_1}\}$ and $f_w(\boldsymbol{x}, \boldsymbol{z}) = \boldsymbol{W}_2(\boldsymbol{z} \odot \sigma(\boldsymbol{W}_1\boldsymbol{x}))$ where $\sigma$ is 1-Lipschitz activation function, $\odot$ denotes element-wise multiplication and $\boldsymbol{z} \sim \mathcal{N}(\boldsymbol{1}, \beta^2\boldsymbol{I}^{d_1})$. Let $\boldsymbol{a} = \sigma(\boldsymbol{W}_1\boldsymbol{x})$, $\hat{\boldsymbol{a}} = \hat{\mathbb{E}}_i[\boldsymbol{a}^{(i)} \odot \boldsymbol{a}^{(i)}]$, $\hat{a}_k \geq 0$ denotes the $k$-th element of $\hat{\boldsymbol{a}}$. Let $\|\boldsymbol{W}_2\|_{2,1} = \sum_k \|\boldsymbol{w}_k\|_2$ where $\boldsymbol{w}_k$ is the $k$-th column vector in $\boldsymbol{W}_2$. The regression task with the $l_2$ loss leads to the weight regularization as:*

$$\hat{L}_{noise}(w) := \hat{\mathbb{E}}_i\mathbb{E}_{\boldsymbol{z}}\big[\|\boldsymbol{y}^{(i)} - \boldsymbol{W}_2(\boldsymbol{z} \odot \boldsymbol{a}^{(i)})\|_2^2\big] = \hat{\mathbb{E}}_i\big[\|\boldsymbol{y}^{(i)} - \boldsymbol{W}_2\boldsymbol{a}^{(i)}\|_2^2\big] + \beta^2\sum_k \hat{a}_k\|\boldsymbol{w}_k\|_2^2. \tag{5}$$

The proof can be found in §C.1. Theorem 1 reveals that modulation with Gaussian multiplicative noise in the regression task implicitly regularizes the internal weight norm, with the regularization strength determined by the variance of the noise $\beta^2$ and the magnitude of the features.

Despite analysis in a simplified two-layer system, the implicit weight regularization by the noise modulation applies to multi-layer and convolutional neural networks, which can be expressed as a combination of two-layer nets, and convolutional layers are a type of linear layer [37, 60, 3]. Assuming latent features in deep neural networks follow a Gaussian distribution [34], we can equate the Bernoulli noise modulation (Dropout) with $\boldsymbol{Z} \sim \mathcal{B}(\beta)$, whose values are set to be $1/(1-\beta)$ with probability $1 - \beta$ and 0 otherwise, to the Gaussian multiplicative noise modulation in subsequent layers, making the regularization effects from Theorem 1 applicable to the Bernoulli noise as well.

> Theorem 1 illustrates that one can modulate the latent features in the discriminator by the multiplicative noise to adaptively regularize the norm of its weights, leading to the reduced Rademacher complexity in Lemma 2 and improved GANs generalization in Lemma 1.

## 3.3 Consistency regularization

Although the noise modulation can reduce the Rademacher complexity, incorporating the noise increases the gradient norms of latent features and the inputs to the discriminator, making training of GAN unstable [56] and difficult to converge [36]. To understand the gradient-related challenges in the noise-modulated discriminator, we adopt the Taylor expansion with a focus on the pivotal first- and second-order terms, following standard practice [1, 12, 66].

**Proposition 1** *Define $f := f_t \circ \ldots \circ f_2$ as the feature extractor of the discriminator from the second layer onward. Let $\tilde{a} = f_1(\tilde{x})$ and $a = f_1(x)$ be the latent features of the first layer for fake and real images, respectively. The discriminator is defined as $h(\cdot) = \phi(\varphi(f(\cdot)))$, with $\varphi$ as the prediction head and $\phi$ as the loss function. Introduce a multiplicative noise $z \sim \mathcal{N}(1, \beta^2 I^{d'})$ modulating solely the first layer. Let $H_{kk}^{(h)}(a)$ be the $k$-th diagonal entry of the Hessian matrix of $h$ at $a$ and $a_k$ be the $k$-th element of $a$. Applying Taylor expansion to $h(\cdot)$, the GAN loss can be expressed as follows:*

$$\min_{\theta_d} L_D^{AN} := \mathbb{E}_{\tilde{a}}\mathbb{E}_z\big[h(z \odot \tilde{a})\big] - \mathbb{E}_a\mathbb{E}_z\big[h(z \odot a)\big]$$

$$\approx \mathbb{E}_{\tilde{a}}\big[h(\tilde{a})\big] - \mathbb{E}_a\big[h(a)\big] + \tfrac{\beta^2}{2}\big(\mathbb{E}_{\tilde{a}}\big[\textstyle\sum_k \tilde{a}_k^2 H_{kk}^{(h)}(\tilde{a})\big] - \mathbb{E}_a\big[\textstyle\sum_k a_k^2 H_{kk}^{(h)}(a)\big]\big), \quad (6)$$

$$\min_{\theta_g} L_G^{AN} := -\mathbb{E}_z\mathbb{E}_{\tilde{a}}\big[h(z \odot \tilde{a})\big] \approx -\mathbb{E}_{\tilde{a}}\big[h(\tilde{a})\big] - \tfrac{\beta^2}{2}\mathbb{E}_{\tilde{a}}\big[\textstyle\sum_k \tilde{a}_k^2 H_{kk}^{(h)}(\tilde{a})\big]. \quad (7)$$

> Prop. 1 (the proof is in §C.2) shows that introducing the noise modulation during the training of the discriminator amplifies the value of diagonal entry of Hessian matrix at real images, causing abrupt changes in their gradient and potentially an increase of the gradient norm of real images. Eq. 7 further shows that as the generator is optimized to produce images resembling real images, the gradient norm of the fake images also experiences amplification. Consequently, gradient norms for both real and fake images increase. Empirically, we observe an increase in the gradient norm of latent features before the prediction head, resulting in a larger gradient norm of samples.

To mitigate this problem, we propose a consistency regularization for $f$, promoting invariance of $f$ to the same input under varying noise modulations. For simplicity, we take $f : \mathbb{R}^{d'} \to \mathbb{R}$, noting that this analysis readily extends to the consistency loss for $f$ with a multidimensional output.

**Theorem 2** *Using notations from Prop. 1 and with $f : \mathbb{R}^{d'} \to \mathbb{R}$, let $\nabla_k^2 f(a)$ and $\big(H_{jk}^{(f)}(a)\big)^2$ denote the squares of the $k$-th gradient element and $(j,k)$-th entry of Hessian matrix of $f$ at $a$, respectively. Given $z_1, z_2 \sim \mathcal{N}(1, \beta^2 I^{d'})$, enforcing invariance of $f$ under different noise modulations yields:*

$$\ell^{NICE}(a) := \mathbb{E}_{z_1, z_2}\big[\big(f(z_1 \odot a) - f(z_2 \odot a)\big)^2\big] \approx 2\beta^2\textstyle\sum_k a_k^2 \nabla_k^2 f(a) + \beta^4\textstyle\sum_{j,k} a_j^2 a_k^2 (H_{jk}^{(f)}(a))^2. \quad (8)$$

> Theorem 2 (the proof is in §C.3) shows that the consistency loss essentially penalizes the first- and second-order gradient of $f$ at $a$, with the strength adaptively controlled by the noise variance $\beta^2$ and modulated by $a$. This results in reweighted gradient squared norms modulation, with terms in expressions $\|\nabla f(a)\|_2^2 = \sum_k \nabla_k^2 f(a)$, and $\|H^{(f)}(a)\|_F^2 = \sum_{j,k}(H_{jk}^{(f)}(a))^2$ influenced by $a_k^2 \geq 0$. In contrast to the noise-modulated discriminator in Eq. 6, NICE regularizes the first- and second-order gradients of layers ahead of the classification head. Despite the gradient from the classification head is large, NICE ensures the gradient reaching the input stays small, resulting in a stable training and improved generalization and convergence [4, 26, 67].

## 3.4 Implementation of NICE

The analysis above highlights practical ways to enhance the generalization and stability of GANs. Firstly, we apply the adaptive multiplicative noise to the latent features of the discriminator, striking balance between discrimination and generalization, as highlighted in [67]. The consistency regularization is then applied to ensure invariance of the discriminator to the same inputs under varied noise modulations, penalizing the gradient and stabilizing the training process.

**GAN with the adaptive noise modulation (AN).** Figure 1 shows the discriminator composed of blocks $B_1, B_2, \ldots, B_L$, where each block $B_l$ contains multiple convolution weights $c \in \{C_1, C_2, C_S\}$. To achieve a trade-off between generalization and discrimination, we modulate the latent features $X \in \mathbb{R}^{1 \times d' \times d^H \times d^W}$ from convolutional layers of the discriminator using adaptive noise $Z \in \mathbb{R}^{1 \times d'}$ by:

$$AN(X) = X \odot Z, \quad (9)$$

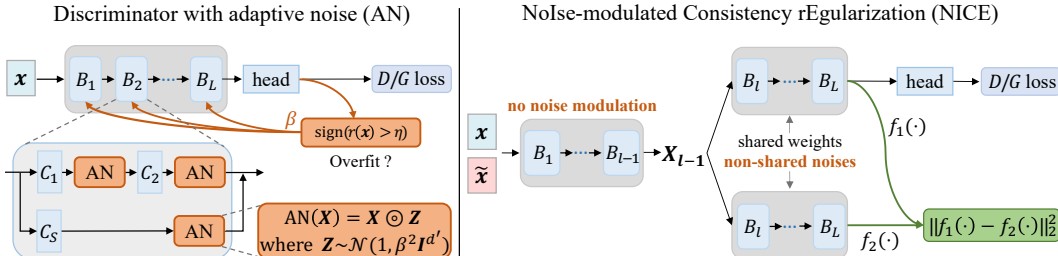

Figure 1: Our discriminator pipeline. NICE is applied to both real $\boldsymbol{x}$ and fake $\tilde{\boldsymbol{x}}$ images.

where $\odot$ represents the operation that expands $\boldsymbol{Z}$ into $1 \times d' \times d^H \times d^W$ shape and performs element-wise multiplication with $\boldsymbol{X}$. The noise $\boldsymbol{Z}$ is carefully controlled through an adaptive $\beta$. In the case of BigGAN and OmniGAN, we control the variance of $\boldsymbol{Z}$ using $\beta$ as $\boldsymbol{Z} \sim \mathcal{N}(\mathbf{1}, \beta^2 \boldsymbol{I}^{d'})$. For StyleGAN2, we control the noise through $\boldsymbol{Z} \sim \mathcal{B}(\beta)$, where with probability $1 - \beta$ it takes the value $\frac{1}{1-\beta}$, and with probability $\beta$ it takes 0.

Below we introduce a mechanism to control the noise via the meta-parameter $\beta$ by detecting potential overfitting in the discriminator. Firstly, we compute the expectation over the discriminator output $r(\boldsymbol{x}) = \mathbb{E}[\text{sign}(D(\boldsymbol{x}))]$ w.r.t. real samples $\boldsymbol{x}$, and evaluate $\varepsilon = \text{sign}(r(\boldsymbol{x}) > \eta) \in \{-1, 0, 1\}$ where $\eta$ is a fixed threshold. A value greater than the threshold indicates potential overfitting [22]. We apply $\beta_{t+1} = \beta_t + \Delta_\beta \cdot \varepsilon$ to update $\beta$ with smaller $\Delta_\beta$ to control $\boldsymbol{Z}$. Denote the modified discriminator with our adaptive noise as $h_{\text{AN}}$, the objective of GAN with the adaptive noise modulation becomes:

$$\begin{cases} L_D^{\text{AN}} = \min_{\boldsymbol{\theta}_d} \mathbb{E}_{\tilde{\boldsymbol{x}} \sim \nu_n}[h_{\text{AN}}(\tilde{\boldsymbol{x}}; \boldsymbol{\theta}_d)] - \mathbb{E}_{\boldsymbol{x} \sim \hat{\mu}_n}[h_{\text{AN}}(\boldsymbol{x}; \boldsymbol{\theta}_d)], \\ L_G^{\text{AN}} = \min_{\boldsymbol{\theta}_g} -\mathbb{E}_{\boldsymbol{z} \sim p_z}[h_{\text{AN}}(g(\boldsymbol{z}; \boldsymbol{\theta}_g))]. \end{cases} \quad (10)$$

**GAN with the noise-modulated consistency regularization (NICE).** To deal with the increased gradient norm due to the adaptive noise modulation, we introduce the consistency regularization (NICE) to promote invariance of the discriminator to the same samples under different noise modulations. Our regularization implicitly penalizes the gradient of the latent features in the discriminator by comparing outputs $f_1(\cdot)$ and $f_2(\cdot)$ of two feature extractors (with shared parameters) that are subjected to different noise modulations as in Figure 1. The objective for a GAN with NICE is:

$$\begin{cases} L_D^{\text{NICE}} = \min_{\boldsymbol{\theta}_d} \mathbb{E}_{\tilde{\boldsymbol{x}} \sim \nu_n}[h_{\text{AN}}(\tilde{\boldsymbol{x}}; \boldsymbol{\theta}_d) + \gamma \ell^{\text{NICE}}(\tilde{\boldsymbol{x}})] + \mathbb{E}_{\boldsymbol{x} \sim \hat{\mu}_n}[-h_{\text{AN}}(\boldsymbol{x}; \boldsymbol{\theta}_d) + \gamma \ell^{\text{NICE}}(\boldsymbol{x})], \\ L_G^{\text{NICE}} = \min_{\boldsymbol{\theta}_g} \mathbb{E}_{\boldsymbol{z} \sim p_z}[-h_{\text{AN}}(g(\boldsymbol{z}; \boldsymbol{\theta}_g)) + \gamma \ell^{\text{NICE}}(g(\boldsymbol{z}; \boldsymbol{\theta}_g))], \end{cases} \quad (11)$$

where $\ell^{\text{NICE}}(\cdot) = \|f_1(\cdot) - f_2(\cdot)\|_2^2$ and $\gamma = \Delta_\gamma \beta$ is a meta-parameter that controls the strength of the consistency regularization and is adaptively determined by $\beta$.

**Efficient NICE.** In our experiments, we observed that applying the noise modulation in the later blocks of the discriminator often outperformed applying noise in the earlier blocks. To avoid unnecessary computations (the early blocks do not use modulation), we divide the discriminator into a modulation-free part $(B_1, \ldots, B_{l-1})$ and a noise-modulated part $(B_l, \ldots, B_L)$, where $l$ indicates the starting block of noise modulation. For efficient calculation of the consistency loss, we input two copies of $\boldsymbol{X}_{l-1}$ (output from the first part) into the second part, bypassing the need for redundant calculations to maintain a low computational overhead compared to standard approaches.

## 4 Experiments

We conduct experiments on CIFAR-10/100 [28] using BigGAN [7] and OmniGAN [71], as well as on ImageNet [10] using BigGAN for conditional image generation. We also evaluate our method on low-shot datasets [68], which include 100-shot Obama/Panda/Grumpy Cat and AnimalFace Dog/Cat [51], and FFHQ [24] using StyleGAN2 [25]. We compare our method against several strong baselines, including DA [68], ADA [22], DigGAN [11], MaskedGAN [17], KDDLGAN [9], LeCam [57], GenCo [8], InsGen [63], FakeCLR [32] and TransferGAN [59]. For fair comparison, we denote methods using massive augmentation as "MA", which include DA and ADA.

**Datasets.** CIFAR-10 has 50K/10K training/testing images with resolution of $32 \times 32$ from 10 categories, whereas CIFAR-100 has 100 classes. FFHQ contains 70K human face images at $256 \times 256$

pixels. Low-shot datasets contain 100-shot Obama/Panda/Grumpy Cat images, AnimalFace (160 cats and 389 dogs) images at $256 \times 256$ resolution. ImageNet has 1.2M/50K training/validation images with 1K categories. Following [17, 9], we center-crop and downscale its images to $64 \times 64$ resolution. The implementation details can be found in §D. The generated images can be found in §J.

**Evaluation metrics.** We generate 50K images per dataset to compute the commonly used Inception Score [46] and Fréchet Inception Distance (FID) [15]. We report tFID, computed between 50K generated images and all training images. For CIFAR-10/100, we also compute vFID between 10K generated images and 10K real testing images. For low-shot datasets, we follow [68] and compute FID between 5K generated images and the entire dataset. For FFHQ, we calculate FID between 50K fake images and the entire training set. For ImageNet, we follow [17, 9] and generate 10K images for computing the IS and FID, where the reference distribution is the entire training set. Following [68, 11, 32], we run 5 trails for methods using NICE, and report the mean of the results. Given that all standard deviations fall below the 1% relative, we omit them for clarity.

## 4.1 Results on CIFAR-10 and CIFAR-100 for BigGAN and OmniGAN

Tables 1 and 2 demonstrate that NICE consistently outperforms baselines such as BigGAN, LeCam+DA, OmniGAN and OmniGAN+ADA on CIFAR-10 and CIFAR-100, firmly establishing its superiority. NICE also outperforms LeCam+DA+KDDLGAN in the majority of scenarios without any knowledge integration from large-scale models, underscoring its efficiency and effectiveness.

Table 1: Comparison w/ and w/o NICE on CIFAR-10 given different percentage of training data.

| Method | MA | 100% data | | | 20% data | | | 10% data | | |
| --- | --- | --- | --- | --- | --- | --- | --- | --- | --- | --- |
| | | IS↑ | tFID↓ | vFID↓ | IS↑ | tFID↓ | vFID↓ | IS↑ | tFID↓ | vFID↓ |
| BigGAN($d'=256$) | × | 9.21 | 5.48 | 9.42 | 8.74 | 16.20 | 20.27 | 8.24 | 31.45 | 35.59 |
| +LeCam | × | 9.45 | 4.27 | 8.29 | 8.95 | 11.34 | 15.25 | 8.44 | 28.36 | 33.65 |
| +DigGAN | × | 9.28 | 5.33 | 9.35 | 8.81 | 13.28 | 17.25 | 8.32 | 18.54 | 22.45 |
| +NICE | × | **9.50** | **4.19** | **8.24** | **8.96** | **8.51** | **12.54** | **8.73** | **13.65** | **17.75** |
| +LeCam+DA | ✓ | 9.45 | 4.32 | 8.40 | 9.01 | 8.53 | 12.47 | 8.81 | 12.64 | 16.42 |
| +LeCam+DA+KDDLGAN | ✓ | – | – | 8.19 | – | – | 11.15 | – | – | 13.86 |
| +LeCam+DA+NICE | ✓ | **9.52** | **3.72** | **7.81** | **9.12** | **6.92** | **10.89** | **8.99** | **9.86** | **13.81** |
| OmniGAN($d'=1024$) | × | 10.01 | 6.92 | 10.75 | 8.64 | 36.75 | 41.17 | 6.69 | 53.02 | 57.68 |
| +DA | ✓ | 10.13 | 4.15 | 8.06 | 9.49 | 13.45 | 17.27 | 8.99 | 19.45 | 23.48 |
| +ADA | ✓ | 10.24 | 4.95 | 9.06 | 9.41 | 27.04 | 30.58 | 7.86 | 40.05 | 44.01 |
| +NICE | × | 10.21 | 2.72 | 6.79 | 9.86 | 6.06 | 9.87 | 9.78 | 6.40 | 10.37 |
| +NICE+ADA | ✓ | **10.38** | **2.25** | **6.32** | **10.18** | **4.39** | **8.42** | **10.08** | **5.49** | **9.42** |

Table 2: Comparison w/ and w/o NICE on CIFAR-100 given different percentage of training data.

| Method | MA | 100% data | | | 20% data | | | 10% data | | |
| --- | --- | --- | --- | --- | --- | --- | --- | --- | --- | --- |
| | | IS↑ | tFID↓ | vFID↓ | IS↑ | tFID↓ | vFID↓ | IS↑ | tFID↓ | vFID↓ |
| BigGAN($d'=256$) | × | 11.02 | 7.86 | 12.70 | 9.94 | 25.83 | 30.79 | 7.58 | 50.79 | 55.04 |
| +LeCam | × | **11.41** | 6.82 | 11.54 | 10.05 | 20.81 | 25.77 | 8.14 | 41.51 | 46.43 |
| +DigGAN | × | 11.15 | 8.13 | 13.06 | 9.98 | 16.87 | 21.59 | **9.04** | 23.10 | 27.78 |
| +NICE | × | 10.99 | **6.31** | **11.08** | **10.32** | **13.17** | **17.80** | 8.96 | **19.53** | **24.33** |
| +LeCam+DA | ✓ | 11.25 | 6.45 | 11.26 | 10.12 | 15.96 | 20.42 | 9.17 | 22.75 | 27.14 |
| +LeCam+DA+KDDLGAN | ✓ | – | – | **10.12** | – | – | 18.70 | – | – | 22.40 |
| +LeCam+DA+NICE | ✓ | **11.28** | **5.72** | 10.40 | **10.54** | **10.02** | **14.93** | **9.35** | **14.95** | **19.60** |
| OmniGAN($d'=1024$) | × | 12.73 | 8.36 | 13.18 | 10.14 | 40.59 | 44.92 | 6.91 | 60.46 | 64.76 |
| +DA | ✓ | 12.94 | 7.41 | 12.08 | 11.35 | 17.65 | 22.37 | 10.01 | 30.68 | 34.94 |
| +ADA | ✓ | 13.07 | 6.12 | 10.79 | 12.07 | 13.54 | 18.20 | 8.95 | 44.65 | 49.08 |
| +NICE | × | 13.77 | 3.83 | 8.61 | 12.57 | 8.68 | 13.53 | 11.97 | 14.53 | 19.22 |
| +NICE+ADA | ✓ | **13.82** | **3.78** | **8.59** | **12.75** | **6.28** | **10.92** | **12.04** | **9.32** | **14.18** |

## 4.2 Results on low-shot generation on StyleGAN2

Table 3 showcases the superior performance of NICE in comparison to leading methods, significantly improving baselines and achieving state-of-the-art results. NICE also surpasses KDDLGAN, despite KDDLGAN leverages the large-scale CLIP model [43]. Figure 2 shows realistic images generated by NICE under scarce data training.

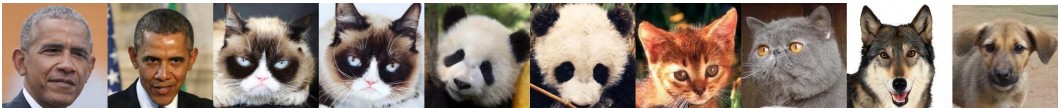

Figure 2: Images generated using NICE+ADA on StyleGAN2; see Figure 14 in §J for more examples and comparisons for this low-shot generation setting.

Table 3: FID ↓ scores for unconditional image generation with StyleGAN2 on 100-shot Obama/Grumpy cat/Panda and AnimalFace-Cat/Dog datasets. † indicates using a generator pre-trained on the full FFHQ dataset. ‡ means using the CLIP [43] model pre-trained on large scale data.

| Method | MA | Pre-trained | 100-shot | | | Animal Face | |
| --- | --- | --- | --- | --- | --- | --- | --- |
| | | | Obama | Grumpy Cat | Panda | Cat | Dog |
| StyleGAN2 [25] | ✗ | ✗ | 80.20 | 48.90 | 34.27 | 71.71 | 131.90 |
| StyleGAN2+SSGAN-LA [16] | ✗ | ✗ | 79.88 | 38.42 | 28.6 | 78.78 | 109.91 |
| StyleGAN2+NICE | ✗ | ✗ | **24.56** | **18.78** | **8.92** | **25.25** | **46.56** |
| ADA [22] | ✓ | ✗ | 45.69 | 26.62 | 12.90 | 40.77 | 56.83 |
| DA [68] | ✓ | ✗ | 46.87 | 27.08 | 12.06 | 42.44 | 58.85 |
| DigGAN [11] | ✓ | ✗ | 36.38 | 25.42 | 11.54 | 35.67 | 59.98 |
| LeCam [57] | ✓ | ✗ | 33.16 | 24.93 | 10.16 | 34.18 | 54.88 |
| GenCo [8] | ✓ | ✗ | 32.21 | 17.79 | 9.49 | 30.89 | 49.63 |
| InsGen [63] | ✓ | ✗ | 32.42 | 22.01 | 9.85 | 33.01 | 44.93 |
| MaskedGAN [17] | ✓ | ✗ | 33.78 | 20.06 | 8.93 | – | – |
| FakeCLR [32] | ✓ | ✗ | 26.95 | 19.56 | 8.42 | 26.34 | 42.02 |
| TransferGAN † [59] | ✓ | ✓ | 39.85 | 29.77 | 17.12 | 49.10 | 65.57 |
| LeCam+KDDLGAN ‡ [9] | ✓ | ✓ | 29.38 | 19.65 | 8.41 | 31.89 | 50.22 |
| ADA+NICE | ✓ | ✗ | **20.09** | **15.63** | **8.18** | **22.70** | **28.65** |

## 4.3 Results on FFHQ for StyleGAN2 and ImageNet for BigGAN

Results for unconditional image generation on FFHQ are in Table 4, and for conditional image generation on ImageNet $64 \times 64$ in Table 5. We limit FFHQ to 100, 1K, 2K, 5K real images and follow [17, 9] for ImageNet setting. NICE showcases superior performance on FFHQ and ImageNet.

Table 4: FID ↓ scores on FFHQ using StyleGAN2. ADA-Linear is introduced in [63].

| Method | MA | FFHQ | | | |
| --- | --- | --- | --- | --- | --- |
| | | 100 | 1K | 2K | 5K |
| StyleGAN2 | ✗ | 179 | 100.16 | 54 | 49.68 |
| ADA | ✓ | 85.8 | 21.29 | 15.39 | 10.96 |
| ADA-Linear | ✓ | 82 | 19.86 | 13.01 | 9.39 |
| InsGen | ✓ | 45.75 | 18.21 | 11.47 | 7.83 |
| FakeCLR | ✓ | 42.56 | 15.92 | 9.90 | 7.25 |
| ADA+NICE | ✓ | **38.42** | **14.57** | **8.85** | **6.48** |

Table 5: Comparison with and w/o NICE on ImageNet given different percentage of training data.

| Method | MA | 10% data | | 5% data | | 2.5% data | |
| --- | --- | --- | --- | --- | --- | --- | --- |
| | | IS ↑ | FID ↓ | IS ↑ | FID ↓ | IS↑ | FID ↓ |
| BigGAN | ✗ | 10.94 | 38.30 | 6.13 | 91.16 | 3.92 | 133.80 |
| ADA | ✓ | 12.67 | 31.89 | 9.44 | 43.21 | 8.54 | 56.83 |
| DA | ✓ | 12.76 | 32.82 | 9.63 | 56.75 | 8.17 | 63.49 |
| MaskedGAN | ✓ | 13.34 | 26.51 | 12.85 | 35.70 | 12.68 | 38.62 |
| KDDLGAN | ✓ | 14.14 | 20.32 | 14.06 | 22.35 | **14.65** | 28.79 |
| NICE | ✗ | 14.18 | 21.44 | 13.96 | 24.72 | 13.32 | 31.45 |
| ADA+NICE | ✓ | **14.58** | **18.29** | **14.10** | **20.07** | 13.92 | **24.41** |

## 4.4 Analysis of NICE

**Analysis of the stabilizing effect of NICE.** In our 10% CIFAR-10 experiments with OmniGAN($d' = 256$), Figure 3 provides compelling evidence supporting our theory. Figure 3a shows that Omni-GAN+AN and OmniGAN+NICE achieve lower weight norms than OmniGAN, validating Theorem

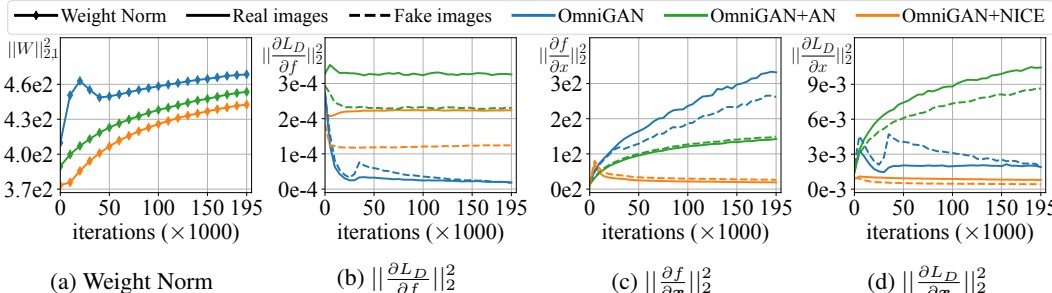

Figure 3: Weight and gradient norms of the discriminator on 10% CIFAR-10 with OmniGAN ($d' = 256$). (a) total discriminator weight norms, (b) gradient norm at the layer before classification head, (c) gradient norm of $f$ w.r.t. input, and (d) gradient norm of discriminator loss w.r.t. input.

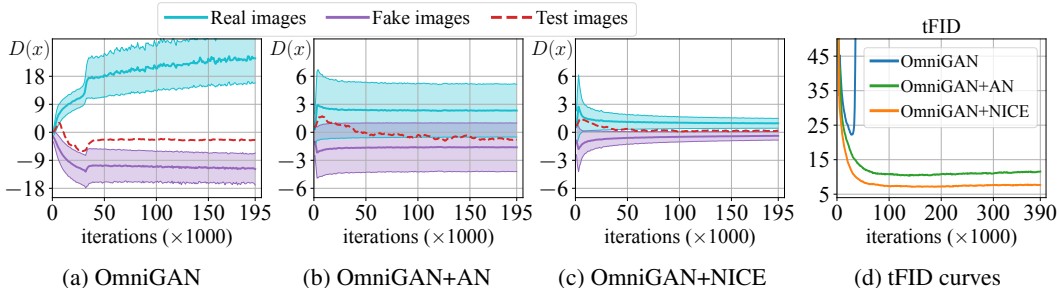

Figure 4: The discriminator output w.r.t. real, fake and test images of (a) OmniGAN, (b) Omni-GAN+AN, (c) OmniGAN+NICE, along with (d) tFID curves on 10% CIFAR-10 using OmniGAN ($d' = 256$). The shaded region represents the standard deviation. Note we scale the $y$-axis in (b) and (c) for visual clarity. In (d), training time is doubled to evaluate the endurance of our methods under prolonged training conditions.

1 that the multiplicative noise modulation reduces the weight norm of discriminator, thus lowering the Rademacher complexity and improving generalization.

Figure 3b shows a gradient surge at the latent layer before the classification head, especially for real images in OmniGAN+AN and OmniGAN+NICE, surpassing gradients of OmniGAN. This observation aligns with our prediction in Prop. 1 that introducing noise amplifies gradients due to the maximization effect on real images when training discriminator and on fake images when training generator, causing undesired gradient issues.

Figure 3c illustrates a smaller squared gradient norm $\|\partial f / \partial \boldsymbol{x}\|_2^2$ for OmniGAN+NICE compared to OmniGAN+AN and OmniGAN, showcasing the effectiveness of consistency regularization in penalizing the gradient of discriminator, in line with the theoretical derivation in Theorem 2.

Figure 3d demonstrates that OmniGAN+NICE achieves a lower gradient norm of the discriminator loss w.r.t. the input than both OmniGAN and OmniGAN+AN, affirming ability of NICE to counteract the negative effects of large gradient norms caused by noise modulation. The resulted smaller gradients at the input validates efficiency of NICE in stabilizing training.

**Analysis of the ability of NICE to enhance generalization.** Figure 4 visualizes the discriminator outputs for real, fake, and test images along with tFID curves. In contrast to OmniGAN, Omni-GAN+AN and OmniGAN+NICE balance discrimination and generalization [67], maintaining a steady discrepancy between real and fake images. This equilibrium ensures a smaller discrepancy between real seen images and unseen samples, effectively lowering the generation error as outlined in Eq. 3 of Lemma 1. As a result, the generalization of GANs is enhanced, leading to superior performance for image generation, as illustrated by the tFID curves in Figure 4d.

## 4.5 Ablation studies

**Ablation of components in NICE.** We examine the impact of various components of NICE: (i) introducing adaptive noise (AN) to the discriminator, (ii) applying consistency regularization to

Table 6: Ablation studies on components (AN, $\text{NICE}_{D_r}$, $\text{NICE}_{D_f}$, $\text{NICE}_{G_f}$) in NICE.

| Method | | | | 10% CIFAR-10 | | | 10% CIFAR-100 | | | Obama |
|---|---|---|---|---|---|---|---|---|---|---|
| AN | $\text{NICE}_{D_r}$ | $\text{NICE}_{D_f}$ | $\text{NICE}_{G_f}$ | IS↑ | tFID↓ | vFID↓ | IS↑ | tFID↓ | vFID↓ | FID↓ |
| | | | | 8.49 | 22.24 | 26.33 | 8.19 | 45.41 | 50.33 | 80.26 |
| ✓ | | | | 9.16 | 10.14 | 13.80 | 11.22 | 23.76 | 28.34 | 44.68 |
| ✓ | ✓ | | | 9.17 | 8.88 | 12.69 | 11.38 | 20.09 | 24.62 | 32.14 |
| ✓ | ✓ | | ✓ | 9.16 | 8.69 | 12.59 | 11.19 | 18.80 | 24.13 | 29.95 |
| ✓ | ✓ | ✓ | | 9.23 | 7.45 | 11.25 | **11.62** | 18.73 | 23.42 | 27.93 |
| ✓ | ✓ | ✓ | ✓ | **9.26** | **7.23** | **11.08** | 11.50 | **16.91** | **21.56** | **24.56** |

real images ($\text{NICE}_{D_r}$), (iii) applying consistency regularization to fake images ($\text{NICE}_{D_f}$) during discriminator training, and (iv) enforcing consistency on fake images while training the generator ($\text{NICE}_{G_f}$). Table 6 shows our evaluations across 10% CIFAR-10/100 using OmniGAN ($d' = 256$) and the Obama on StyleGAN2. The optimal performance is achieved when NICE is applied to both real and fake images in the training of both the discriminator and generator.

**Impact of different factors in NICE.** For an in-depth analysis on how different factors in NICE contribute to its performance, readers are directed to §E.

**Comparison with alternatives for enhanced generalization and stability.** Table 7 shows that NICE outperforms DA, ADA, AWD (adaptive weight decay), AN+AGP (adaptive noise with adaptive gradient penalization) and $\text{NICE}_{add}$ (with additive noise). Detailed implementation of these variants can be found in §D.2. Figure 5 shows that NICE also enhances performance with increasing network size, while ADA and DA exhibit a decline. As augmentation-based methods such as DA, ADA, and AACR, may leak augmentation cues to the generator (Figure 9 of §I), NICE mitigates this drawback.

| Method | 10% CIFAR-10 | | | 10% CIFAR-100 | | |
|---|---|---|---|---|---|---|
| | IS↑ | tFID↓ | vFID↓ | IS↑ | tFID↓ | vFID↓ |
| OmniGAN | 8.49 | 22.24 | 26.33 | 8.14 | 45.41 | 50.33 |
| +AWD | 8.56 | 18.28 | 22.12 | 9.64 | 37.68 | 42.01 |
| +AN+AGP | 8.98 | 11.55 | 15.25 | 10.72 | 27.73 | 32.15 |
| +$\text{NICE}_{add}$ | 8.64 | 17.94 | 21.59 | 9.34 | 28.59 | 33.02 |
| +DA | 8.84 | 12.90 | 16.67 | 10.16 | 24.50 | 28.96 |
| +ADA | **9.67** | 13.86 | 17.70 | 11.23 | 23.11 | 27.58 |
| +AACR | 9.63 | 10.93 | 14.73 | 11.37 | 21.42 | 25.76 |
| +NICE | 9.26 | **7.23** | **11.08** | **11.50** | **16.91** | **21.56** |

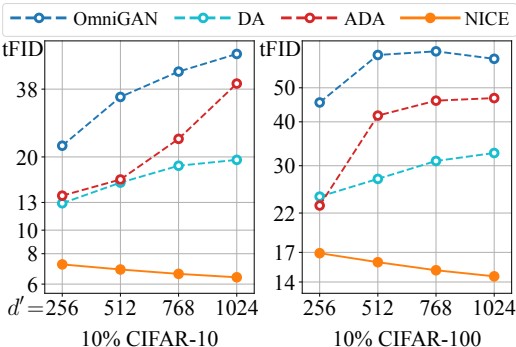

Table 7: Various methods+OmniGAN ($d' = 256$). Figure 5: tFID ↓ w.r.t. different network sizes $d'$.

Table 7 & Fig. 5: Various methods on OmniGAN. Tab. 7: AWD is Adaptive weight decay. AN+AGP: Adaptive noise + adaptive gradient penalty. $\text{NICE}_{add}$: NICE with additive noise. AACR: Adaptive augmentation-based consistency reg. Fig. 5: tFID ↓ w.r.t. different $d'$ on 10% CIFAR-10/100.

**Rationalizing Advantages of NICE.** Refer to §F and §G for a comprehensive explanations of why NICE outperforms other alternatives, enhancing generalization and stability. Given a mild increase in the computational load (§H), substantial performance gains are achieved.

# 5 Conclusions

Our newly proposed approach for GAN training, NoIse-modulated Consistency rEgularization (NICE), improves generalization of various GAN models and their performance under limited data training. Through rigorous theoretical analysis, we have shown that NICE reduces the Rademacher complexity and penalizes the gradient of the latent features within the discriminator. Our experimental results match theoretical findings, illustrating that NICE not only improves generalization but also enhances stability of GAN training. By harnessing these two key advantages, NICE works with various network backbones and consistently outperforms existing methods. These exceptional results firmly establish NICE as a powerful solution for preventing the pervasive issue of discriminator overfitting in GANs. Limitations and border impact of our work are discussed in §A.

## Acknowledgements

We thank to Moyang Liu, whose discussions and constant encouragement significantly shaped this work. YN is supported by China Scholarship Council (202008520034). PK is supported by CSIRO's Science Digital. We are grateful for the valuable feedback provided by all anonymous reviewers.

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

# NICE: NoIse-modulated Consistency rEgularization for Data-Efficient GANs (Supplementary Material)

**Yao Ni**[†], **Piotr Koniusz**[*,§,†]
†The Australian National University    §Data61♥CSIRO
†firstname.lastname@anu.edu.au

## A  Broader impact and limitations

NICE is a solution to limiting the extensive image usage in GAN training. Conventional GANs heavily rely on massive training data, which can jeopardize data privacy during large-scale data collection. NICE is an innovative method for data-limited image generation. By successfully training GANs with just 100 images, NICE not only minimizes the dependency on large-scale training datasets, but also significantly reduces the training cost. NICE decreases the risk of privacy violations (using smaller datasets equals to the lesser risk of data mismanagement). NICE also provides a cost-effective alternative for training GANs. As the data required for training is lesser compared with large-scale models, NICE is also energy-consumption friendly. Of course, with small-scale dataset comes a limitation on diversity. The model cannot generate instances for which the data is completely lacking. Thus, special care needs to be taken regarding gender, race and other biases in generated images.

NICE advances GANs with Theorems 1 and 2, along with Corollaries 1 and 2. It offers novel insights into the weight regularization. These findings, driven by the multiplicative and additive noise modulation, along with the gradient regularization due to the consistency regularization, pave the way for further future research.

## B  Notations

Below, we explain the notations used in this work.

**Scalars**: Represented by lowercase letters (*e.g.*, $m$, $n$, $\delta$).

**Vectors**: Bold lowercase letters (*e.g.*, $\boldsymbol{v}$, $\boldsymbol{u}$, $\boldsymbol{m}$).

**Matrices**: Bold uppercase letters (*e.g.*, $\boldsymbol{W}$, $\boldsymbol{X}$, $\boldsymbol{H}$).

**Vector/Matrix elements**: $v_k$ and $a_k$ represent $k$-th element of $\boldsymbol{v}$ and $\boldsymbol{a}$. $H_{jk}$ denotes the element located at the $j$-th row and $k$-th column of matrix $\boldsymbol{H}$.

**Functions**: Letters followed by brackets (*e.g.*, $\sigma(\cdot)$, $h(\cdot)$, $\nabla f(\cdot)$, $H(\cdot)$).

**Function sets**: Calligraphic uppercase letters are used, but note that $\mathcal{N}$ and $\mathcal{B}$ specifically denote Gaussian and Bernoulli distributions, respectively (*e.g.*, $\mathcal{H}$, $\mathcal{F}$).

**Probability measures**: Denoted by letters $\mu$, $\nu$ and $p_z$.

**Dataset samples**: Expressed as $\boldsymbol{x}^{(i)}$, $\boldsymbol{y}^{(i)}$, $\boldsymbol{a}^{(i)}$, $\tau^{(i)}$.

**Multiplication/Composition**: $\odot$ represents element-wise multiplication and $\circ$ denotes function composition.

**Expectation/empirical expectation**: $\mathbb{E}[\cdot]$ represents the average or expected value of a random variable while $\hat{\mathbb{E}}[\cdot]$ denotes the empirical expectation calculated over observed data samples.

---

*The corresponding author.  Code: `https://github.com/MaxwellYaoNi/NICE`

# C Proofs

We define $\|\boldsymbol{v}\|^2 = \boldsymbol{v}^T\boldsymbol{v}$ as the squared norm of vector $\boldsymbol{v}$. Note the commutative property of the dot product, $\boldsymbol{v}^T\boldsymbol{u} = \boldsymbol{u}^T\boldsymbol{v}$, and the distributive property over matrices, $\boldsymbol{v}^T\boldsymbol{A} + \boldsymbol{u}^T\boldsymbol{A} = (\boldsymbol{v} + \boldsymbol{u})^T\boldsymbol{A}$. With these definitions and properties established, we now prove Theorem 1, Proposition 1 and Theorem 2. We begin with a simplified scalar-based derivation, ensuring a rapid derivation of the final results, followed by comprehensive proofs encompassing vector and matrix calculus for a holistic understanding.

## C.1 Proof of Theorem 1

**Theorem 1** *(**Regularization by the Gaussian multiplicative noise in deep regression**) Consider a regression task on a two-layer neural network. Let $\mathcal{X} \subseteq \mathbb{R}^{d_0}$ and $\mathcal{Y} \subseteq [-1,1]^{d_2}$ be the input and output spaces, and $\mathcal{D} = \mathcal{X} \times \mathcal{Y}$. Let $n$ examples $\{(\boldsymbol{x}^{(i)}, \boldsymbol{y}^{(i)})\}_{i=1}^n \sim \mathcal{D}^n$. Let $f_w : \mathcal{X} \to \mathcal{Y}$ be parameterized by $\{w : \boldsymbol{W}_1 \in \mathbb{R}^{d_1 \times d_0}, \boldsymbol{W}_2 \in \mathbb{R}^{d_2 \times d_1}\}$ and $f_w(\boldsymbol{x}, \boldsymbol{z}) = \boldsymbol{W}_2(\boldsymbol{z} \odot \sigma(\boldsymbol{W}_1\boldsymbol{x}))$ where $\sigma$ is 1-Lipschitz activation function, $\odot$ denotes element-wise multiplication and $\boldsymbol{z} \sim \mathcal{N}(\boldsymbol{1}, \beta^2\boldsymbol{I}^{d_1})$. Let $\boldsymbol{a} = \sigma(\boldsymbol{W}_1\boldsymbol{x})$, $\hat{\boldsymbol{a}} = \hat{\mathbb{E}}_i[\boldsymbol{a}^{(i)} \odot \boldsymbol{a}^{(i)}]$, $\hat{a}_k \geq 0$ denotes the k-th element of $\hat{\boldsymbol{a}}$. Let $\|\boldsymbol{W}_2\|_{2,1} = \sum_k \|\boldsymbol{w}_k\|_2$ where $\boldsymbol{w}_k$ is the k-th column vector in $\boldsymbol{W}_2$. The regression task with the $l_2$ loss leads to the weight regularization as:*

$$\hat{L}_{noise}(w) := \hat{\mathbb{E}}_i\mathbb{E}_{\boldsymbol{z}}\big[\|\boldsymbol{y}^{(i)} - \boldsymbol{W}_2(\boldsymbol{z} \odot \boldsymbol{a}^{(i)})\|_2^2\big] = \hat{\mathbb{E}}_i\big[\|\boldsymbol{y}^{(i)} - \boldsymbol{W}_2\boldsymbol{a}^{(i)}\|_2^2\big] + \beta^2\sum_k\hat{a}_k\|\boldsymbol{w}_k\|_2^2.$$

We commence with a simplified derivation using scalars, which can quickly lead to the final results:

$$\hat{\mathbb{E}}_i\mathbb{E}_{z\sim\mathcal{N}(1,\beta^2)}\big[\big(y^{(i)} - w_2(za^{(i)})\big)^2\big] = \hat{\mathbb{E}}_i\mathbb{E}_z\big[\big(y^{(i)} - w_2a^{(i)} - w_2(z-1)a^{(i)}\big)^2\big]$$
$$= \hat{\mathbb{E}}_i\mathbb{E}_z\big[\big(y^{(i)} - w_2a^{(i)}\big)^2 - 2(y^{(i)} - w_2a^{(i)})w_2(z-1)a^{(i)} + \big(w_2(z-1)a^{(i)}\big)^2\big]$$
$$= \hat{\mathbb{E}}_i\big[\big(y^{(i)} - w_2a^{(i)}\big)^2\big] + w_2^2\hat{\mathbb{E}}_ia^{(i)2}\mathbb{E}_z[(z-1)^2]$$
$$= \hat{\mathbb{E}}_i\big[\big(y^{(i)} - w_2a^{(i)}\big)^2\big] + \beta^2w_2^2\hat{\mathbb{E}}_ia^{(i)2}.$$

***Proof:***

$$\hat{L}_{\text{noise}}(w) := \hat{\mathbb{E}}_i\mathbb{E}_{\boldsymbol{z}}\big[\|\boldsymbol{y}^{(i)} - \boldsymbol{W}_2(\boldsymbol{z} \odot \boldsymbol{a}^{(i)})\|_2^2\big] = \hat{\mathbb{E}}_i\mathbb{E}_{\boldsymbol{z}}\big[\|\boldsymbol{y}^{(i)} - \boldsymbol{W}_2\boldsymbol{a}^{(i)} - \boldsymbol{W}_2\big((\boldsymbol{z}-\boldsymbol{1}) \odot \boldsymbol{a}^{(i)}\big)\|_2^2\big]$$
$$= \hat{\mathbb{E}}_i\big[\|\boldsymbol{y}^{(i)} - \boldsymbol{W}_2\boldsymbol{a}^{(i)}\|_2^2\big] - 2\hat{\mathbb{E}}_i\big[(\boldsymbol{y}^{(i)} - \boldsymbol{W}_2\boldsymbol{a}^{(i)})^T\boldsymbol{W}_2\big(\mathbb{E}_{\boldsymbol{z}}[\boldsymbol{z}-\boldsymbol{1}] \odot \boldsymbol{a}^{(i)}\big)\big] + \hat{\mathbb{E}}_i\mathbb{E}_{\boldsymbol{z}}\big[\|\boldsymbol{W}_2\big((\boldsymbol{z}-\boldsymbol{1}) \odot \boldsymbol{a}^{(i)}\big)\|_2^2\big]$$
$$= \hat{\mathbb{E}}_i\big[\|\boldsymbol{y}^{(i)} - \boldsymbol{W}_2\boldsymbol{a}^{(i)}\|_2^2\big] - 2\hat{\mathbb{E}}_i\big[(\boldsymbol{y}^{(i)} - \boldsymbol{W}_2\boldsymbol{a}^{(i)})^T\boldsymbol{W}_2(\boldsymbol{0} \odot \boldsymbol{a}^{(i)})\big] + \hat{\mathbb{E}}_i\mathbb{E}_{\boldsymbol{z}}\big[\|\boldsymbol{W}_2\big((\boldsymbol{z}-\boldsymbol{1}) \odot \boldsymbol{a}^{(i)}\big)\|_2^2\big]$$
$$= \hat{\mathbb{E}}_i\|\boldsymbol{y}^{(i)} - \boldsymbol{W}_2\boldsymbol{a}^{(i)}\|_2^2 + \hat{\mathbb{E}}_i\mathbb{E}_{\boldsymbol{z}}\|\boldsymbol{W}_2\big((\boldsymbol{z}-\boldsymbol{1}) \odot \boldsymbol{a}^{(i)}\big)\|_2^2. \tag{12}$$

Let $\boldsymbol{m} \sim \mathcal{N}(\boldsymbol{0}, \beta^2\boldsymbol{I}^{d_1})$, with $m_k$ and $a_k^{(i)}$ representing the k-th elements of vectors $\boldsymbol{m}$ and $\boldsymbol{a}^{(i)}$, respectively. Using these definitions, we derive the second term of Eq. 12 as follows:

$$\hat{\mathbb{E}}_i\mathbb{E}_{\boldsymbol{z}}\big[\|\boldsymbol{W}_2\big((\boldsymbol{z}-\boldsymbol{1}) \odot \boldsymbol{a}^{(i)}\big)\|_2^2\big] = \hat{\mathbb{E}}_i\mathbb{E}_{\boldsymbol{m}}\big[\|\boldsymbol{W}_2\big(\boldsymbol{m} \odot \boldsymbol{a}^{(i)}\big)\|_2^2\big]$$
$$= \hat{\mathbb{E}}_i\mathbb{E}_{\boldsymbol{m}}\big[(\boldsymbol{m} \odot \boldsymbol{a}^{(i)})^T\boldsymbol{W}_2^T\boldsymbol{W}_2(\boldsymbol{m} \odot \boldsymbol{a}^{(i)})\big]$$
$$= \hat{\mathbb{E}}_i\mathbb{E}_{\boldsymbol{m}}\Big[\sum_{j=1}^{d_1}\sum_{k=1}^{d_1}(m_ja_j^{(i)})(\boldsymbol{W}_2^T\boldsymbol{W}_2)_{jk}(m_ka_k^{(i)})\Big]. \tag{13}$$

Given that the elements in $\boldsymbol{m}$ are independent, for $j \neq k$ we have:

$$\mathbb{E}_{\boldsymbol{m}}\big[(m_ja_j^{(i)})(\boldsymbol{W}_2^T\boldsymbol{W}_2)_{jk}(m_ka_k^{(i)})\big] = 0. \tag{14}$$

Observe that $(\boldsymbol{W}_2^T\boldsymbol{W}_2)_{kk} = \|\boldsymbol{w}_k\|_2^2$ represents the squared norm of k-th column vector in $\boldsymbol{W}_2$. Let $\hat{\boldsymbol{a}}$ denote the element-wise mean square of $\boldsymbol{a}^{(i)}$, *i.e.*, $\hat{\boldsymbol{a}} = \hat{\mathbb{E}}_i[\boldsymbol{a}^{(i)} \odot \boldsymbol{a}^{(i)}]$. We leverage the distribution property $\mathbb{E}[z^2] = \sigma^2 + \mu^2$ for $z \sim \mathcal{N}(\mu, \sigma^2)$, simplifying to $\mathbb{E}[z^2] = \sigma^2$ when $\mu = 0$. This lets us

reformulate Eq. 13 explicitly as:

$$\hat{\mathbb{E}}_i\mathbb{E}_{\boldsymbol{m}}\Big[\sum_{j=1}^{d_1}\sum_{k=1}^{d_1}(m_ja_j^{(i)})(\boldsymbol{W}_2^T\boldsymbol{W}_2)_{jk}(m_ka_k^{(i)})\Big]$$

$$=\hat{\mathbb{E}}_i\mathbb{E}_{\boldsymbol{m}}\Big[\sum_{k=1}^{d_1}(m_ka_k^{(i)})(\boldsymbol{W}_2^T\boldsymbol{W}_2)_{kk}(m_ka_k^{(i)})\Big]$$

$$=\hat{\mathbb{E}}_i\mathbb{E}_{\boldsymbol{m}}\sum_{k=1}^{d_1}(m_k^2a_k^{(i)2})\|\boldsymbol{w}_k\|_2^2=\beta^2\hat{\mathbb{E}}_i\sum_{k=1}^{d_1}a_k^{(i)2}\|\boldsymbol{w}_k\|_2^2$$

$$=\beta^2\sum_k\hat{a}_k\|\boldsymbol{w}_k\|_2^2. \tag{15}$$

Substituting the second term in Eq. 12 with Eq. 15, we arrive at:

$$\hat{L}_{\text{noise}}(w):=\hat{\mathbb{E}}_i\mathbb{E}_{\boldsymbol{z}}\big[\|\boldsymbol{y}^{(i)}-\boldsymbol{W}_2(\boldsymbol{z}\odot\boldsymbol{a}^{(i)})\|_2^2\big]=\hat{\mathbb{E}}_i\big[\|\boldsymbol{y}^{(i)}-\boldsymbol{W}_2\boldsymbol{a}^{(i)}\|_2^2\big]+\beta^2\sum_k\hat{a}_k\|\boldsymbol{w}_k\|_2^2. \tag{16}$$

Integrating the Gaussian multiplicative noise into the optimization of regression task leads to an adaptive regularization $\beta^2\sum_k\hat{a}_k\|\boldsymbol{w}_k\|_2^2$ on $\boldsymbol{W}_2$, effectively reweighting the squared norm $\|\boldsymbol{w}_k\|_2^2$ of each column in accordance with $\hat{a}_k\geq0$. This mechanism, driven by the variance $\beta^2$ of noise enhances the robustness of model and its generalization capability. It is worth highlighting that this process relates closely to the (2,1)-norm of $\boldsymbol{W}_2$, defined as $\|\boldsymbol{W}_2\|_{2,1}=\sum_k\|\boldsymbol{w}_k\|_2$.

**Corollary 1** *Utilizing the additive noise injection where $\boldsymbol{z}\sim\mathcal{N}(\boldsymbol{0},\beta^2\boldsymbol{I}^{d_1})$, the regression task with the $l_2$ loss yields weight regularization as:*

$$\hat{L}_{add\_noise}(w):=\hat{\mathbb{E}}_i\mathbb{E}_{\boldsymbol{z}}\big[\|\boldsymbol{y}^{(i)}-\boldsymbol{W}_2(\boldsymbol{z}+\boldsymbol{a}^{(i)})\|_2^2\big]=\hat{\mathbb{E}}_i\big[\|\boldsymbol{y}^{(i)}-\boldsymbol{W}_2\boldsymbol{a}^{(i)}\|_2^2\big]+\beta^2\sum_k\|\boldsymbol{w}_k\|_2^2. \tag{17}$$

## C.2   Proof of Proposition 1

**Proposition 1** *Define $f:=f_t\circ\ldots\circ f_2$ as the feature extractor of the discriminator from the second layer onward. Let $\tilde{\boldsymbol{a}}=f_1(\tilde{\boldsymbol{x}})$ and $\boldsymbol{a}=f_1(\boldsymbol{x})$ be the latent features of the first layer for fake and real images, respectively. The discriminator is defined as $h(\cdot)=\phi(\varphi(f(\cdot)))$, with $\varphi$ as the prediction head and $\phi$ as the loss function. Introduce a multiplicative noise $\boldsymbol{z}\sim\mathcal{N}(\boldsymbol{1},\beta^2\boldsymbol{I}^{d'})$ modulating solely the first layer. Let $H_{kk}^{(h)}(\boldsymbol{a})$ be the $k$-th diagonal entry of the Hessian matrix of $h$ at $\boldsymbol{a}$ and $a_k$ be the $k$-th element of $\boldsymbol{a}$. Applying Taylor expansion to $h(\cdot)$, the GAN loss can be expressed as follows:*

$$\min_{\boldsymbol{\theta}_d}L_D^{AN}:=\mathbb{E}_{\tilde{\boldsymbol{a}}}\mathbb{E}_{\boldsymbol{z}}\big[h(\boldsymbol{z}\odot\tilde{\boldsymbol{a}})\big]-\mathbb{E}_{\boldsymbol{a}}\mathbb{E}_{\boldsymbol{z}}\big[h(\boldsymbol{z}\odot\boldsymbol{a})\big]$$

$$\approx\mathbb{E}_{\tilde{\boldsymbol{a}}}\big[h(\tilde{\boldsymbol{a}})\big]-\mathbb{E}_{\boldsymbol{a}}\big[h(\boldsymbol{a})\big]+\tfrac{\beta^2}{2}\big(\mathbb{E}_{\tilde{\boldsymbol{a}}}\big[\sum_k\tilde{a}_k^2H_{kk}^{(h)}(\tilde{\boldsymbol{a}})\big]-\mathbb{E}_{\boldsymbol{a}}\big[\sum_ka_k^2H_{kk}^{(h)}(\boldsymbol{a})\big]\big).$$

$$\min_{\boldsymbol{\theta}_g}L_G^{AN}:=-\mathbb{E}_{\boldsymbol{z}}\mathbb{E}_{\tilde{\boldsymbol{a}}}\big[h(\boldsymbol{z}\odot\tilde{\boldsymbol{a}})\big]\approx-\mathbb{E}_{\tilde{\boldsymbol{a}}}\big[h(\tilde{\boldsymbol{a}})\big]-\tfrac{\beta^2}{2}\mathbb{E}_{\tilde{\boldsymbol{a}}}\big[\sum_k\tilde{a}_k^2H_{kk}^{(h)}(\tilde{\boldsymbol{a}})\big].$$

We provide a succinct proof for the discriminator using scalar values to quickly arrive at the final result, where $h'(\cdot)$ and $h''(\cdot)$ are the first- and second-order derivatives, respectively.

$$\mathbb{E}_zh(z\tilde{a})-\mathbb{E}_zh(za)=\mathbb{E}_zh\big(\tilde{a}+(z-1)\tilde{a}\big)-\mathbb{E}_zh\big(a+(z-1)a\big)$$

$$=\mathbb{E}_z\big[h(\tilde{a})+(z-1)\tilde{a}h'(\tilde{a})+\frac{(z-1)^2}{2}\tilde{a}^2h''(\tilde{a})\big]-\mathbb{E}_z\big[h(a)+(z-1)ah'(a)+\frac{(z-1)^2}{2}a^2h''(a)\big]$$

$$=h(\tilde{a})-h(a)+\mathbb{E}_z\big[\frac{(z-1)^2}{2}\big]\big(\tilde{a}^2h''(\tilde{a})-a^2h''(a)\big)$$

$$=h(\tilde{a})-h(a)+\frac{\beta^2}{2}\big(\tilde{a}^2h''(\tilde{a})-a^2h''(a)\big).$$

**Proof:**   Consider a multivariate normal distribution $\boldsymbol{m}\sim\mathcal{N}(\boldsymbol{0},\beta^2\boldsymbol{I}^{d'})$. Let $\nabla h(\boldsymbol{a})$ and $H_{jk}^{(h)}(\boldsymbol{a})$ represent the first-order derivative and the $(j,k)$-th entry of the Hessian matrix of $h$ at $\boldsymbol{a}$, respectively,

implying that $H_{jk}^{(h)} = H_{kj}^{(h)}$ due to the symmetry of the Hessian matrix. With these notations established, we proceed as follows:

$$
\begin{aligned}
L_D^{\mathrm{AN}} &:= \mathbb{E}_{\tilde{\boldsymbol{a}}}\mathbb{E}_{\boldsymbol{z}}\big[h(\tilde{\boldsymbol{a}} \odot \boldsymbol{z})\big] - \mathbb{E}_{\boldsymbol{a}}\mathbb{E}_{\boldsymbol{z}}\big[h(\boldsymbol{a} \odot \boldsymbol{z})\big] \\
&= \mathbb{E}_{\tilde{\boldsymbol{a}}}\mathbb{E}_{\boldsymbol{z}}\big[h\big(\tilde{\boldsymbol{a}} + (\boldsymbol{z} - \boldsymbol{1}) \odot \tilde{\boldsymbol{a}}\big)\big] - \mathbb{E}_{\boldsymbol{a}}\mathbb{E}_{\boldsymbol{z}}\big[h\big(\boldsymbol{a} + (\boldsymbol{z} - \boldsymbol{1}) \odot \boldsymbol{a}\big)\big] \\
&= \mathbb{E}_{\tilde{\boldsymbol{a}}}\mathbb{E}_{\boldsymbol{m}}\big[h(\tilde{\boldsymbol{a}} + \boldsymbol{m} \odot \tilde{\boldsymbol{a}})\big] - \mathbb{E}_{\boldsymbol{a}}\mathbb{E}_{\boldsymbol{m}}\big[h(\boldsymbol{a} + (\boldsymbol{m} \odot \boldsymbol{a}))\big] \\
&\approx \mathbb{E}_{\tilde{\boldsymbol{a}}}\mathbb{E}_{\boldsymbol{m}}\Big[h(\tilde{\boldsymbol{a}}) + (\boldsymbol{m} \odot \tilde{\boldsymbol{a}})^T \nabla h(\tilde{\boldsymbol{a}}) + \frac{1}{2}(\boldsymbol{m} \odot \tilde{\boldsymbol{a}})^T H^{(h)}(\tilde{\boldsymbol{a}})(\boldsymbol{m} \odot \tilde{\boldsymbol{a}})\Big] \\
&\quad - \mathbb{E}_{\boldsymbol{a}}\mathbb{E}_{\boldsymbol{m}}\Big[h(\boldsymbol{a}) + (\boldsymbol{m} \odot \boldsymbol{a})^T \nabla h(\boldsymbol{a}) + \frac{1}{2}(\boldsymbol{m} \odot \boldsymbol{a})^T H^{(h)}(\boldsymbol{a})(\boldsymbol{m} \odot \boldsymbol{a})\Big] \\
&= \mathbb{E}_{\tilde{\boldsymbol{a}}}h(\tilde{\boldsymbol{a}}) - \mathbb{E}_{\boldsymbol{a}}h(\boldsymbol{a}) \\
&\quad + \mathbb{E}_{\tilde{\boldsymbol{a}}}\mathbb{E}_{\boldsymbol{m}}\big[(\boldsymbol{m} \odot \tilde{\boldsymbol{a}})^T \nabla h(\tilde{\boldsymbol{a}})\big] - \mathbb{E}_{\boldsymbol{a}}\mathbb{E}_{\boldsymbol{m}}\big[(\boldsymbol{m} \odot \boldsymbol{a})^T \nabla h(\boldsymbol{a})\big] \qquad (18) \\
&\quad + \mathbb{E}_{\tilde{\boldsymbol{a}}}\mathbb{E}_{\boldsymbol{m}}\big[\tfrac{1}{2}(\boldsymbol{m} \odot \tilde{\boldsymbol{a}})^T H^{(h)}(\tilde{\boldsymbol{a}})(\boldsymbol{m} \odot \tilde{\boldsymbol{a}})\big] - \mathbb{E}_{\boldsymbol{a}}\mathbb{E}_{\boldsymbol{m}}\tfrac{1}{2}(\boldsymbol{m} \odot \boldsymbol{a})^T H^{(h)}(\boldsymbol{a})(\boldsymbol{m} \odot \boldsymbol{a}). \qquad (19)
\end{aligned}
$$

Addressing the first-order gradient term presented in Eq. 18, we notice the following:

$$
\begin{aligned}
&\mathbb{E}_{\tilde{\boldsymbol{a}}}\mathbb{E}_{\boldsymbol{m}}\big[(\boldsymbol{m} \odot \tilde{\boldsymbol{a}})^T \nabla h(\tilde{\boldsymbol{a}})\big] - \mathbb{E}_{\boldsymbol{a}}\mathbb{E}_{\boldsymbol{m}}\big[(\boldsymbol{m} \odot \boldsymbol{a})^T \nabla h(\boldsymbol{a})\big] \\
=&\mathbb{E}_{\tilde{\boldsymbol{a}}}\big[(\boldsymbol{0} \odot \tilde{\boldsymbol{a}})^T \nabla h(\tilde{\boldsymbol{a}})\big] - \mathbb{E}_{\boldsymbol{a}}\big[(\boldsymbol{0} \odot \boldsymbol{a})^T \nabla h(\boldsymbol{a})\big] = 0. \qquad (20)
\end{aligned}
$$

Examining the second-order gradient in Eq. 19, and leveraging the independence of elements in $\boldsymbol{m}$, we apply the result from Eq. 14. With $\tilde{a}_k$ and $a_k$ representing the $k$-th elements in $\tilde{\boldsymbol{a}}$ and $\boldsymbol{a}$, respectively, we derive:

$$
\begin{aligned}
&\mathbb{E}_{\tilde{\boldsymbol{a}}}\mathbb{E}_{\boldsymbol{m}}\big[\tfrac{1}{2}(\boldsymbol{m} \odot \tilde{\boldsymbol{a}})^T H^{(h)}(\tilde{\boldsymbol{a}})(\boldsymbol{m} \odot \tilde{\boldsymbol{a}})\big] - \mathbb{E}_{\boldsymbol{a}}\mathbb{E}_{\boldsymbol{m}}\big[\tfrac{1}{2}(\boldsymbol{m} \odot \boldsymbol{a})^T H^{(h)}(\boldsymbol{a})(\boldsymbol{m} \odot \boldsymbol{a})\big] \\
=&\mathbb{E}_{\tilde{\boldsymbol{a}}}\mathbb{E}_{\boldsymbol{m}}\Big[\frac{1}{2}\sum_{k=1}^{d'} m_k \tilde{a}_k H_{kk}^{(h)}(\tilde{\boldsymbol{a}}) m_k \tilde{a}_k\Big] - \mathbb{E}_{\boldsymbol{a}}\mathbb{E}_{\boldsymbol{m}}\Big[\frac{1}{2}\sum_{k=1}^{d'} m_k a_k H_{kk}^{(h)}(\boldsymbol{a}) m_k a_k\Big] \\
=&\frac{\beta^2}{2}\Big(\mathbb{E}_{\tilde{\boldsymbol{a}}}\Big[\sum_k \tilde{a}_k^2 H_{kk}^{(h)}(\tilde{\boldsymbol{a}})\Big] - \mathbb{E}_{\boldsymbol{a}}\Big[\sum_k a_k^2 H_{kk}^{(h)}(\boldsymbol{a})\Big]\Big). \qquad (21)
\end{aligned}
$$

By substituting Eq. 18 with the results from Eq. 20, and Eq. 19 with Eq. 21, we arrive at the following expression:

$$
L_D^{\mathrm{AN}} \approx \mathbb{E}_{\tilde{\boldsymbol{a}}}\big[h(\tilde{\boldsymbol{a}})\big] - \mathbb{E}_{\boldsymbol{a}}\big[h(\boldsymbol{a})\big] + \frac{\beta^2}{2}\Big(\mathbb{E}_{\tilde{\boldsymbol{a}}}\Big[\sum_k \tilde{a}_k^2 H_{kk}^{(h)}(\tilde{\boldsymbol{a}})\Big] - \mathbb{E}_{\boldsymbol{a}}\Big[\sum_k a_k^2 H_{kk}^{(h)}(\boldsymbol{a})\Big]\Big). \qquad (22)
$$

Utilizing the same derivation process as before for $L_G^{\mathrm{AN}}$, we arrive at the following result:

$$
L_G^{\mathrm{AN}} \approx -\mathbb{E}_{\tilde{\boldsymbol{a}}}\big[h(\tilde{\boldsymbol{a}})\big] - \frac{\beta^2}{2}\mathbb{E}_{\tilde{\boldsymbol{a}}}\Big[\sum_k \tilde{a}_k^2 H_{kk}^{(h)}(\tilde{\boldsymbol{a}})\Big]. \qquad (23)
$$

Introducing the multiplicative Gaussian noise during training of GAN produces the gradient maximization effect on the real samples in the discriminator training, specially through the term $\frac{\beta^2}{2}\sum_k a_k^2 H_{kk}^{(h)}(\boldsymbol{a})$. This term intertwines the second-order partial derivative (diagonal entry $H_{kk}^{(h)}(\boldsymbol{a})$ of the Hessian matrix) with the non-negative factor $\frac{\beta^2}{2}a_k^2 \geq 0$, and is related to the trace of the Hessian matrix given by $\mathbf{trace}\big(H^{(h)}(\boldsymbol{a}_r)\big) = \sum_k H_{kk}^{(h)}(\boldsymbol{a})$. Employing similar reasoning for the fake samples when training the generator reveals that the noise-modulated discriminator training of GAN tends to amplify the the diagonal entry of the second-order partial derivative in the Hessian matrix at both the real and fake points, influencing the optimization stability.

### C.3  Proof of Theorem 2

**Theorem 2** *Using notations from Prop. 1 and with $f: \mathbb{R}^{d'} \rightarrow \mathbb{R}$, let $\nabla_k^2 f(\boldsymbol{a})$ and $\big(H_{jk}^{(f)}(\boldsymbol{a})\big)^2$ denote the squares of the $k$-th gradient element and $(j,k)$-th entry of Hessian matrix of $f$ at $\boldsymbol{a}$, respectively. Given $\boldsymbol{z}_1, \boldsymbol{z}_2 \sim \mathcal{N}(\boldsymbol{1}, \beta^2 \boldsymbol{I}^{d'})$, enforcing invariance of $f$ under different noise modulations yields:*

$$
\ell^{NICE}(\boldsymbol{a}) := \mathbb{E}_{\boldsymbol{z}_1, \boldsymbol{z}_2}\big[\big(f(\boldsymbol{z}_1 \odot \boldsymbol{a}) - f(\boldsymbol{z}_2 \odot \boldsymbol{a})\big)^2\big] \approx 2\beta^2 \sum_k a_k^2 \nabla_k^2 f(\boldsymbol{a}) + \beta^4 \sum_{j,k} a_j^2 a_k^2 (H_{jk}^{(f)}(\boldsymbol{a}))^2 .
$$

A concise derivation employs scalar values for the Taylor expansion, swiftly leading to results using $f'(\cdot)$ and $f''(\cdot)$ for first and second-order derivatives, respectively.

$$
\mathbb{E}_{z_1,z_2}\big(f(az_1) - f(az_2)\big)^2 = \mathbb{E}_{z_1,z_2}\big(f(a + (z_1-1)a) - f(a + (z_2-1)a)\big)^2
$$
$$
=\mathbb{E}_{z_1,z_2}\Big(f(a)+(z_1-1)af'(a)+\frac{(z_1-1)^2}{2}a^2f''(a)-f(a)-(z_2-1)af'(a)-\frac{(z_2-1)^2}{2}a^2f''(a)\Big)^2
$$
$$
=\mathbb{E}_{z_1,z_2}\Big((z_1 - z_2)af'(a) + \frac{(z_1-1)^2 - (z_2-1)^2}{2}a^2f''(a)\Big)^2
$$
$$
=\mathbb{E}_{m_1,m_2\sim\mathcal{N}(0,\beta^2)}\Big((m_1 - m_2)af'(a) + \frac{m_1^2 - m_2^2}{2}a^2f''(a)\Big)^2
$$
$$
=\mathbb{E}_{m_1,m_2}\Big[(m_1 - m_2)^2a^2f'^2(a) + (m_1 - m_2)(m_1^2 - m_2^2)a^3f'(a)f''(a) + \frac{(m_1^2 - m_2^2)^2}{4}a^4f''^2(a)\Big]
$$
$$
=2\beta^2a^2f'^2(a) + \beta^4a^4f''^2(a).
$$

***Proof:*** Let $m_1, m_2 \sim \mathcal{N}(0, \beta^2\boldsymbol{I}^{d'})$. Note $z_1$ and $z_2$ are independent of each other, as well as independent of $\boldsymbol{a}$ and $f(\boldsymbol{a})$. The same independence holds for $z_2$. By taking expectations over $z_1$ and $z_2$, and applying Taylor expansion, we analyze the consistency loss of $f(\boldsymbol{a})$:

$$
\ell^{\text{NICE}}(\boldsymbol{a}) :=\mathbb{E}_{\boldsymbol{z}_1,\boldsymbol{z}_2}\big[\big(f(\boldsymbol{a}\odot\boldsymbol{z}_1) - f(\boldsymbol{a}\odot\boldsymbol{z}_2)\big)^2\big]
$$
$$
=\mathbb{E}_{\boldsymbol{m}_1,\boldsymbol{m}_2}\Big[\big(f(\boldsymbol{a}+\boldsymbol{m}_1\odot\boldsymbol{a}) - f(\boldsymbol{a}+\boldsymbol{m}_2\odot\boldsymbol{a})\big)^2\Big]
$$
$$
\approx\mathbb{E}_{\boldsymbol{m}_1,\boldsymbol{m}_2}\Big[\big(f(\boldsymbol{a}) + (\boldsymbol{m}_1\odot\boldsymbol{a})^T\nabla f(\boldsymbol{a}) + \frac{1}{2}(\boldsymbol{m}_1\odot\boldsymbol{a})^TH^{(f)}(\boldsymbol{a})(\boldsymbol{m}_1\odot\boldsymbol{a})
$$
$$
- f(\boldsymbol{a}) - (\boldsymbol{m}_2\odot\boldsymbol{a})^T\nabla f(\boldsymbol{a}) - \frac{1}{2}(\boldsymbol{m}_2\odot\boldsymbol{a})^TH^{(f)}(\boldsymbol{a})(\boldsymbol{m}_2\odot\boldsymbol{a})\big)^2\Big]
$$
$$
=\mathbb{E}_{\boldsymbol{m}_1,\boldsymbol{m}_2}\Big[\big((\boldsymbol{m}_1\odot\boldsymbol{a})^T\nabla f(\boldsymbol{a}) + \frac{1}{2}(\boldsymbol{m}_1\odot\boldsymbol{a})^TH^{(f)}(\boldsymbol{a})(\boldsymbol{m}_1\odot\boldsymbol{a})
$$
$$
- (\boldsymbol{m}_2\odot\boldsymbol{a})^T\nabla f(\boldsymbol{a}) - \frac{1}{2}(\boldsymbol{m}_2\odot\boldsymbol{a})^TH^{(f)}(\boldsymbol{a})(\boldsymbol{m}_2\odot\boldsymbol{a})\big)^2\Big]. \tag{24}
$$

Defining $\boldsymbol{v} := \boldsymbol{m}_1 \odot \boldsymbol{a}$ and $\boldsymbol{u} := \boldsymbol{m}_2 \odot \boldsymbol{a}$ with $\boldsymbol{m}_1$ and $\boldsymbol{m}_2$ are independently sampled from $\mathcal{N}(\boldsymbol{0}, \beta^2\boldsymbol{I}^{d'})$, we get $\boldsymbol{v}, \boldsymbol{u} \sim \mathcal{N}(\boldsymbol{0}, \beta^2\text{diag}(\boldsymbol{a}\odot\boldsymbol{a}))$. Here $\text{diag}(\boldsymbol{a}\odot\boldsymbol{a})$ is a diagonal matrix with $a_k^2$ on the diagonal. Letting $\boldsymbol{\delta} := \nabla f(\boldsymbol{a})$ and $\boldsymbol{H} = H^{(f)}(\boldsymbol{a})$, we can rewrite Eq. 24 as follows:

$$
\mathbb{E}_{\boldsymbol{v},\boldsymbol{u}}\big[\big(\boldsymbol{v}^T\boldsymbol{\delta} + \frac{1}{2}\boldsymbol{v}^T\boldsymbol{H}\boldsymbol{v} - \boldsymbol{u}^T\boldsymbol{\delta} - \frac{1}{2}\boldsymbol{u}^T\boldsymbol{H}\boldsymbol{u}\big)^2\big]
$$
$$
=\mathbb{E}_{\boldsymbol{v},\boldsymbol{u}}\big[\big((\boldsymbol{v}-\boldsymbol{u})^T\boldsymbol{\delta} + \frac{1}{2}\boldsymbol{v}^T\boldsymbol{H}\boldsymbol{v} - \frac{1}{2}\boldsymbol{u}^T\boldsymbol{H}\boldsymbol{u}\big)^2\big]
$$
$$
=\mathbb{E}_{\boldsymbol{v},\boldsymbol{u}}\Big[\big((\boldsymbol{v}-\boldsymbol{u})^T\boldsymbol{\delta}\big)^2\Big] \tag{25}
$$
$$
+ \mathbb{E}_{\boldsymbol{v},\boldsymbol{u}}\Big[\big((\boldsymbol{v}-\boldsymbol{u})^T\boldsymbol{\delta}\big)\big(\boldsymbol{v}^T\boldsymbol{H}\boldsymbol{v} - \boldsymbol{u}^T\boldsymbol{H}\boldsymbol{u}\big)\Big] \tag{26}
$$
$$
+ \frac{1}{4}\mathbb{E}_{\boldsymbol{v}}\big[(\boldsymbol{v}^T\boldsymbol{H}\boldsymbol{v})^2\big] + \frac{1}{4}\mathbb{E}_{\boldsymbol{u}}\big[(\boldsymbol{u}^T\boldsymbol{H}\boldsymbol{u})^2\big] \tag{27}
$$
$$
- \frac{1}{2}\mathbb{E}_{\boldsymbol{v},\boldsymbol{u}}\big[(\boldsymbol{v}^T\boldsymbol{H}\boldsymbol{v})(\boldsymbol{u}^T\boldsymbol{H}\boldsymbol{u})\big]. \tag{28}
$$

Next, we succinctly derive the four terms: the first (Eq. 25), second (Eq. 26), third (Eq. 27), and fourth (Eq. 28) term, respectively.

**The first term (Eq. 25):**

Leveraging the property $\mathbb{E}_{z_1,z_2}[(z_1 - z_2)^2] = 2\sigma^2$ for $z_1, z_2 \sim \mathcal{N}(0, \sigma^2)$, we can simplify (Eq. 25) as follows:

$$
\mathbb{E}_{\boldsymbol{v},\boldsymbol{u}}\Big[\big((\boldsymbol{v}-\boldsymbol{u})^T\boldsymbol{\delta}\big)^2\Big] = \mathbb{E}_{\boldsymbol{v},\boldsymbol{u}}\Big[\sum_{k=1}^{d'}(\boldsymbol{v}-\boldsymbol{u})_k^2\delta_k^2\Big] = \sum_{k=1}^{d'}\mathbb{E}_{\boldsymbol{v},\boldsymbol{u}}\big[(\boldsymbol{v}-\boldsymbol{u})_k^2\big]\delta_k^2 = 2\beta^2\sum_k a_k^2\delta_k^2. \tag{29}
$$

**The second term (Eq. 26):**

Utilizing $E[z^3] = \mu^3 + 3\mu\sigma^2$ for $z \sim \mathcal{N}(\mu, \sigma^2)$, and noting that $E[z^3] = 0$ when $\mu = 0$ and elements in $\boldsymbol{v}$ are independent, Eq. 26 is derived as:

$$\mathbb{E}_{\boldsymbol{v},\boldsymbol{u}}\Big[\big((\boldsymbol{v} - \boldsymbol{u})^T\boldsymbol{\delta}\big)\big(\boldsymbol{v}^T\boldsymbol{H}\boldsymbol{v} - \boldsymbol{u}^T\boldsymbol{H}\boldsymbol{u}\big)\Big]$$
$$=\mathbb{E}_{\boldsymbol{v}}\big[(\boldsymbol{v}^T\boldsymbol{\delta})(\boldsymbol{v}^T\boldsymbol{H}\boldsymbol{v})\big]+\mathbb{E}_{\boldsymbol{u}}\big[(\boldsymbol{u}^T\boldsymbol{\delta})(\boldsymbol{u}^T\boldsymbol{H}\boldsymbol{u})\big]-\mathbb{E}_{\boldsymbol{v},\boldsymbol{u}}\big[(\boldsymbol{v}^T\boldsymbol{\delta})(\boldsymbol{u}^T\boldsymbol{H}\boldsymbol{u})\big]-\mathbb{E}_{\boldsymbol{v},\boldsymbol{u}}\big[(\boldsymbol{u}^T\boldsymbol{\delta})(\boldsymbol{v}^T\boldsymbol{H}\boldsymbol{v})\big]$$
$$=2\mathbb{E}_{\boldsymbol{v}}\big[(\boldsymbol{v}^T\boldsymbol{\delta})(\boldsymbol{v}^T\boldsymbol{H}\boldsymbol{v})\big] = 2\mathbb{E}_{\boldsymbol{v}}\Big[\sum_k v_k\delta_k v_k H_{kk} v_k\Big] = 2\mathbb{E}_{\boldsymbol{v}}\Big[\sum_j v_k^3\delta_k H_{kk}\Big] = 0. \tag{30}$$

**The third term (Eq. 27):**

$$\frac{1}{4}\mathbb{E}_{\boldsymbol{v}}\big[(\boldsymbol{v}^T\boldsymbol{H}\boldsymbol{v})^2\big] + \frac{1}{4}\mathbb{E}_{\boldsymbol{u}}\big[(\boldsymbol{u}^T\boldsymbol{H}\boldsymbol{u})^2\big]$$
$$=\frac{1}{2}\mathbb{E}_{\boldsymbol{v}}\big[(\boldsymbol{v}^T\boldsymbol{H}\boldsymbol{v})^2\big] = \frac{1}{2}\mathbb{E}_{\boldsymbol{v}}\Big[\sum_j\sum_k\sum_p\sum_q v_j H_{jk} v_k v_p H_{pq} v_q\Big]. \tag{31}$$

Given the independence of elements in $\boldsymbol{v}$, only terms with an element repeated two or four times contribute non-zero results, leading to four distinct, non-overlapping cases. Using $\mathbb{E}[z^2] = \sigma^2 + \mu^2$ and $\mathbb{E}[z^4] = \mu^4 + 6\mu^2\sigma^2 + 3\sigma^4$ for $z \sim \mathcal{N}(\mu, \sigma^2)$, and simplifying to $\mathbb{E}[z^2] = \sigma^2$ and $\mathbb{E}[z^4] = 3\sigma^4$ when $\mu = 0$, we have:

***Case 1:*** $j = k \neq p = q$, the independence of $v_j$ and $v_p$ simplifies our calculation, leading to:

$$\mathbb{E}_{\boldsymbol{v}}\Big[\sum_j\sum_{p\neq j} v_j^2 H_{jj} v_p^2 H_{pp}\Big] = \sum_{j,p\neq j} H_{jj}H_{pp}\mathbb{E}[v_j^2]\mathbb{E}[v_p^2] = \beta^4\sum_{j,k\neq j} H_{jj}H_{kk}a_j^2 a_k^2. \tag{32}$$

***Case 2:*** For $j = p \neq k = q$, given the independence of $v_j$ and $v_k$, we have:

$$\mathbb{E}_{\boldsymbol{v}}\Big[\sum_j\sum_{k\neq j} v_j H_{jk} v_k v_j H_{jk} v_k\Big] = \sum_{j,k\neq j} H_{jk}^2\mathbb{E}[v_j^2]\mathbb{E}[v_k^2] = \beta^4\sum_{j,k\neq j} H_{jk}^2 a_j^2 a_k^2. \tag{33}$$

***Case 3:*** For $j = q \neq k = p$, leveraging the independence of $v_j$ and $v_k$ as well as the symmetry $H_{jk} = H_{kj}$, we obtain:

$$\mathbb{E}_{\boldsymbol{v}}\Big[\sum_j\sum_{k\neq j} \boldsymbol{v}_j H_{jk} v_k v_k H_{kj} v_j\Big] = \sum_{j,k\neq j} H_{jk}^2\mathbb{E}[v_j^2]\mathbb{E}[v_k^2] = \beta^4\sum_{j,k\neq j} H_{jk}^2 a_j^2 a_k^2. \tag{34}$$

***Case 4:*** For $j = q = k = p$, using $\mathbb{E}[z^4] = 3\sigma^4$ where $z \sim \mathcal{N}(0, \sigma^2)$, we deduce:

$$\mathbb{E}_{\boldsymbol{v}}\Big[\sum_j v_j H_{jj} v_j v_j H_{jj} v_j\Big] = \sum_j H_{jj}^2\mathbb{E}[v_j^4] = \beta^4\sum_j 3H_{jj}^2 a_j^4. \tag{35}$$

Combining Cases 1-4 together, we arive at for Eq. 27:

$$\frac{\beta^4}{2}\Big(\sum_j 3H_{jj}^2 a_j^4 + \sum_{j,k\neq j}(H_{jj}H_{kk} + 2H_{jk}^2)a_j^2 a_k^2\Big). \tag{36}$$

**The forth term (Eq. 28):**

$$-\frac{1}{2}\mathbb{E}_{\boldsymbol{v},\boldsymbol{u}}\big[(\boldsymbol{v}^T\boldsymbol{H}\boldsymbol{v})(\boldsymbol{u}^T\boldsymbol{H}\boldsymbol{u})\big]$$
$$=-\frac{1}{2}\Big(\mathbb{E}_{\boldsymbol{v}}\Big[\sum_j H_{jj}v_j^2\Big]\mathbb{E}_{\boldsymbol{u}}\Big[\sum_k H_{kk}v_k^2\Big]\Big)$$
$$=-\frac{1}{2}\Big(\sum_j H_{jj}\mathbb{E}[v_j^2]\Big)\Big(\sum_k H_{kk}\mathbb{E}[v_k^2]\Big)$$
$$=-\frac{\beta^4}{2}\Big(\sum_j H_{jj}^2 a_j^4 + \sum_{j,k\neq j} H_{jj}H_{kk}a_j^2 a_k^2\Big). \tag{37}$$

Substitute Eq. 29, 30, 36, 37 back into Eq. 25, 26, 27, 28, we have the final results:

$$
\begin{aligned}
&\mathbb{E}_{\boldsymbol{z}_1, \boldsymbol{z}_2}\big[\big(f(\boldsymbol{z}_1 \odot \boldsymbol{a}) - f(\boldsymbol{z}_2 \odot \boldsymbol{a})\big)^2\big] \\
&\approx 2\beta^2 \sum_k a_k^2 \delta_k^2 + 0 \\
&\quad + \frac{\beta^4}{2}\Big(\sum_j 3H_{jj}^2 a_j^4 + \sum_{j,k \neq j}(H_{jj}H_{kk} + 2H_{jk}^2)a_j^2 a_k^2 - \sum_j H_{jj}^2 a_j^4 - \sum_{j,k \neq j} H_{jj}H_{kk}a_j^2 a_k^2\Big) \\
&= 2\beta^2 \sum_k a_k^2 \delta_k^2 + \beta^4\Big(\sum_j H_{jj}^2 a_j^4 + \sum_{j,k \neq j} H_{jk}^2 a_j^2 a_k^2\Big) \\
&= 2\beta^2 \sum_k a_k^2 \delta_k^2 + \beta^4 \sum_j \sum_k H_{jk}^2 a_j^2 a_k^2 \\
&= 2\beta^2 \sum_k a_k^2 \nabla_k^2 f(\boldsymbol{a}) + \beta^4 \sum_j \sum_k a_j^2 a_k^2 (H_{jk}^{(f)}(\boldsymbol{a}))^2.
\end{aligned}
\tag{38}
$$

Enforcing the consistency regularization implicitly penalizes the first- and second-order gradients of $f$ w.r.t. $\boldsymbol{a}$, with an adaptive penalty controlled $\beta$ and modulated by $\boldsymbol{a}$. This modulation recalibrates the squared norms of gradient: the squared norm of the first-order gradient, $\|\nabla f(\boldsymbol{a})\|_2^2 = \sum_k \nabla_k^2 f(\boldsymbol{a})$, and the Frobenius norm of the Hessian, $\|H^{(f)}(\boldsymbol{a})\|_F^2 = \sum_j \sum_k (H_{jk}^{(f)}(\boldsymbol{a}))^2$. Specifically, each term is reweighted by $a_k^2 \geq 0$, ensuring that the penalty applied to each gradient component is scaled proportionally to its associated magnitude of feature.

**Corollary 2** *Enforcing invariance of $f$ under different additive noise injections, where $\boldsymbol{z}_1, \boldsymbol{z}_2 \sim \mathcal{N}(\boldsymbol{0}, \beta^2 \boldsymbol{I}^{d'})$ leads to the gradient regularization as:*

$$
\mathbb{E}_{\boldsymbol{z}_1, \boldsymbol{z}_2}\big[\big(f(\boldsymbol{z}_1 + \boldsymbol{a}) - f(\boldsymbol{z}_2 + \boldsymbol{a})\big)^2\big] = 2\beta^2 \sum_k \nabla_k^2 f(\boldsymbol{a}) + \beta^4 \sum_j \sum_k (H_{jk}^{(f)}(\boldsymbol{a}))^2.
\tag{39}
$$

## D   Implementation details for Experiments

Following previous benchmarks [68, 22], we augmented all datasets with the simple $x$-flips augmentation.

### D.1   Network architecture and hyperparameters

**CIFAR-10/100.** We experiment with OmniGAN ($d' = 256, 1024$) and BigGAN ($d' = 256$) with the batch size of 32. We follow [68] and train the OmniGAN and BigGAN for 1K epochs on the full data and 5K epochs on 10%/20% data setting. We equip the discriminator with the adaptive noise modulation after convolution weights $c \in \{C_1, C_2, C_S\}$ at all blocks $l \in \{1, 2, 3, 4\}$. We set $\Delta_\beta = 0.001, \eta = 0.5, \Delta_\gamma = 10$. Features before loss function are used for NICE on BigGAN.

**ImageNet**. We experiment with BigGAN with the batch size of 512. We use a learning rate of 1e-4 for generator and 4e-4 for discriminator. The noise modulation is placed after convolution layers $c \in \{C_1, C_2, C_S\}$ at blocks $l \in \{3, 4, 5\}$, $\Delta_\beta = 0.001, \eta = 0.5, \Delta_\gamma = 5$. Features before loss function are used for NICE.

**Low-shot images.** We build our NICE upon StyleGAN2 with batch size of 64 and train the networks until the discriminator had seen 25M images. We apply noise modulation with Bernoulli noise after convolutions $c \in \{C_1, C_2\}$ at blocks $l \in \{3, 4, 5, 6\}$, $\Delta_\beta = 0.0001, \eta = 0.9, \Delta_\gamma = 0.05$

**FFHQ**. We experiments on StyleGAN2 with batch size of 64 and train the networks until the discriminator had seen 25M images. We apply noise modulation with Bernoulli noise after convolutions $c \in \{C_1, C_2\}$ at blocks $l \in \{3, 4, 5, 6\}$, $\Delta_\beta = 0.0001, \eta = 0.6, \Delta_\gamma = 0.05$

### D.2   Implementation details for AWD, AN+AGP, and AACR

In Figure 5, we compare NICE with three different approaches: adaptive weight decay (AWD), adaptive noise + adaptive gradient penalty (AN+AGP) and adaptive augmentation-based consistency regularization (AACR). For AWD, we dynamically control the weight decay using $\beta\lambda_{\text{WD}}$. We search

the $\lambda_{\text{WD}} \in \{2\text{e-4}, 1\text{e-4}, 5\text{e-5}, 2\text{e-5}, 1\text{e-5}, 5\text{e-6}\}$ and obtain the best result at 5e-5. For AN+AGP, we employ the adaptive noise modulation and also adaptively penalize the gradient of $\left\|\frac{\partial L_D^{\text{AN}}}{\partial \boldsymbol{x}}\right\|_2^2$ or $\left\|\frac{\partial f}{\partial \boldsymbol{x}}\right\|_2^2$ with adaptive strength $\beta \lambda_{\text{GP}}$. We test different values of $\lambda_{\text{GP}} \in \{0.2, 0.5, 1, 2, 5, 10\}$ and obtain the best results with $\lambda_{\text{GP}} = 2$. For AACR, we incorporate the adaptive differentiable augmentation (ADA) to provide two different views for the same images, and enforce the discriminator to be consistent to the two views of the same images. We adaptively control the strength of AACR with $\beta \lambda_{\text{AACR}}$. We search the $\lambda_{\text{AACR}} \in \{0.2, 0.5, 1, 2, 5, 10\}$ and obtain the best result with $\lambda_{\text{AACR}} = 2$.

# E   Impact of various factors

Below, we analyze the importance of different factors in NICE by conducting ablation experiments. The results are presented in Figure 6. In Figure 6a, we observe that placing the noise modulation module after $C_1$ and $C_2$ improves performance, and the best results are obtained when the noise is placed after all convolution layers. Figure 6b indicates that placing the noise at deeper layers leads to better tFID scores, and the best performance is achieved when the noise is applied at all blocks. Figure 6c shows that $\Delta_\gamma = 10$ is a good choice for NICE. Figure 6d illustrates $\eta = 0.5$ produce overall good performance. Figure 6e plots the results for consistency regularization among $M$ feature extractors $f$ modulated with noise. We use $M = 2$ (*i.e.*, $f_1$ and $f_2$) in the main paper. Figure 6e indicates there is no need to use more feature extractors, as $f_1(\cdot)$ and $f_2(\cdot)$ for consistency regularization produce the best results. Figure 6f presents the results for consistency regularization at every $N$ discriminator steps, indicating that regularizing the discriminator during each iteration is preferable, as larger intervals result in worse performance.

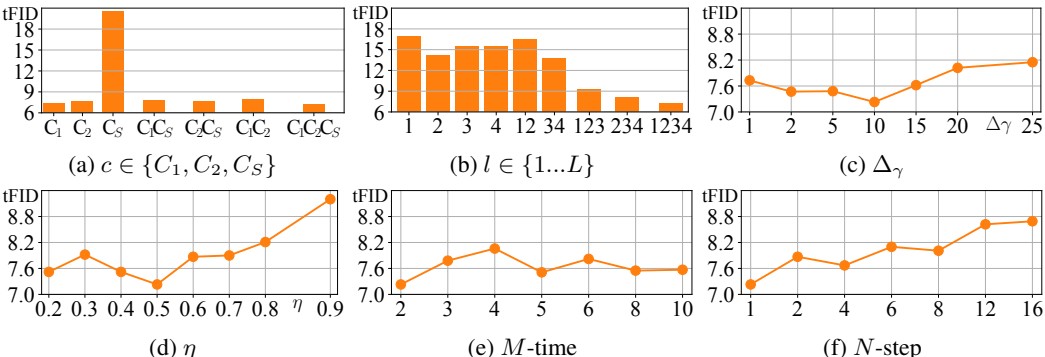

Figure 6: tFID $\downarrow$ under different factors. Ablation studies on 10% CIFAR-10 with OmniGAN ($d' = 256$) w.r.t. different places, different blocks, $\Delta_\gamma$, $\eta$, $M$-time and $N$-step for NICE.

# F   Rationale for choosing multiplicative noise over direct weight regularization

For the multiplicative modulation with the noise, we have:

$$\hat{L}_{\text{mul\_noise}} := \hat{\mathbb{E}}_i \mathbb{E}_{\boldsymbol{z} \sim \mathcal{N}(\boldsymbol{1}, \beta^2 \boldsymbol{I})} \|\boldsymbol{y}^{(i)} - \boldsymbol{W}_2(\boldsymbol{z} \odot \boldsymbol{a}^{(i)})\|_2^2$$
$$= \hat{\mathbb{E}}_i \|\boldsymbol{y}^{(i)} - \boldsymbol{W}_2 \boldsymbol{a}^{(i)}\|_2^2 + \beta^2 \sum_k \hat{a}_k \|\boldsymbol{w}_k\|_2^2,$$

where $\boldsymbol{z} \odot \boldsymbol{a}^{(i)}$ is element-wise multiplication of $\boldsymbol{z}$ with $\boldsymbol{a}^{(i)}$ and $\hat{\boldsymbol{a}} = \hat{\mathbb{E}}_i[\boldsymbol{a}^{(i)} \odot \boldsymbol{a}^{(i)}]$. This implies that while the direct penalty of network weights, $\sum_k \|\boldsymbol{w}_k\|_2^2$, is possible, our approach enjoys a *dynamic modulation* $\beta^2 \hat{a}_k^2$ where variance $\beta^2$ is adapted based on the discriminator decisions and $\hat{a}_k \geq 0$ depends on the magnitude of feature vectors.

> *The importance of such a multiplicative modulation is that semantic contents is not added by noise $\boldsymbol{z}$ to the modulated feature vector $\boldsymbol{z} \cdot \boldsymbol{a}^{(i)}$, i.e.,* only "active" channels $\boldsymbol{a}^{(i)} \neq 0$ are modulated: they can be suppressed or magnified according to variance $\beta^2$.

In contrast, the *additive noise modulation* results in the standard penalty $\beta^2 \sum_k \|\boldsymbol{w}_k\|_2^2$:

$$\hat{L}_{\text{add\_noise}} := \hat{\mathbb{E}}_i \mathbb{E}_{\boldsymbol{z} \sim \mathcal{N}(\boldsymbol{0}, \beta^2 \boldsymbol{I})} \|\boldsymbol{y}^{(i)} - \boldsymbol{W}_2(\boldsymbol{z} + \boldsymbol{a}^{(i)})\|_2^2$$
$$= \hat{\mathbb{E}}_i \|\boldsymbol{y}^{(i)} - \boldsymbol{W}_2 \boldsymbol{a}^{(i)}\|_2^2 + \beta^2 \sum_k \|\boldsymbol{w}_k\|_2^2.$$

> While obvious, $\beta^2 \sum_k \|\boldsymbol{w}_k\|_2^2$ does not enjoy the *dynamic penalty* term related to feature norms. This also means (based on the derivation) that *the additive noise may change feature semantics, i.e.*, $\boldsymbol{z} + \boldsymbol{a}^{(i)}$ may "activate" channels which are non-active in $\boldsymbol{a}^{(i)}$ (features $\boldsymbol{a}^{(i)} = 0$). Therefore, *our multiplicative modulator not only controls the Rademacher Complexity (RC), but controls it in a meaningful manner for discriminator*. Only feature semantics that are present in a feature vector (that describe an object) are modulated while helping control RC. In contrast, the additive noise introduces semantics that are not present for a given image or object.

While directly regularizing the network using $\beta^2 \sum_k \|\boldsymbol{w}_k\|_2^2$ does not introduce the additive noise, the explicit regularization effect is similar to introducing the additive noise. Thus, if we connect the direct weight regularization to the additive noise, it will also have the effect of changing the semantics, which will negatively impact the classification accuracy.

We provide experimental comparisons in Table 8. In addition, Figure 7 shows that AN achives higher classification accuracy than AAN and AWR, demonstrating that multiplicative noise preserves the feature semantics better than the additive noise and the weight regularization.

Table 8: Results for different methods on 10% CIFAR-10/100 using OmniGAN ($d' = 256$). AAN (adaptive additive noise), AWR (adaptive weight regularization), AN (our adaptive multiplicative noise). Please note the consistency loss and dual branch from Figure 1 are not used here as that would result in additional penalties on gradient norms.

| Method | Equation | 10% CIFAR-10 | | | 10% CIFAR-100 | | |
|---|---|---|---|---|---|---|---|
| | | IS ↑ | tFID ↓ | vFID ↓ | IS ↑ | tFID ↓ | vFID ↓ |
| OmniGAN | | 8.49 | 22.24 | 26.33 | 8.19 | 45.41 | 50.33 |
| +AAN | $\boldsymbol{a}^{(i)} := \boldsymbol{a}^{(i)} + \boldsymbol{z}; \boldsymbol{z} \sim \mathcal{N}(\boldsymbol{0}, \beta^2 \boldsymbol{I})$ | 8.52 | 20.12 | 24.65 | 9.64 | 37.68 | 42.01 |
| +AWR | $\beta^2 \sum_k \|\boldsymbol{w}_k\|_2^2$ | 8.44 | 18.42 | 22.56 | 9.80 | 32.05 | 36.53 |
| +AN | $\boldsymbol{a}^{(i)} := \boldsymbol{a}^{(i)} \odot \boldsymbol{z}; \boldsymbol{z} \sim \mathcal{N}(\boldsymbol{1}, \beta^2 \boldsymbol{I})$ | **9.16** | **10.14** | **13.80** | **11.22** | **23.76** | **28.34** |

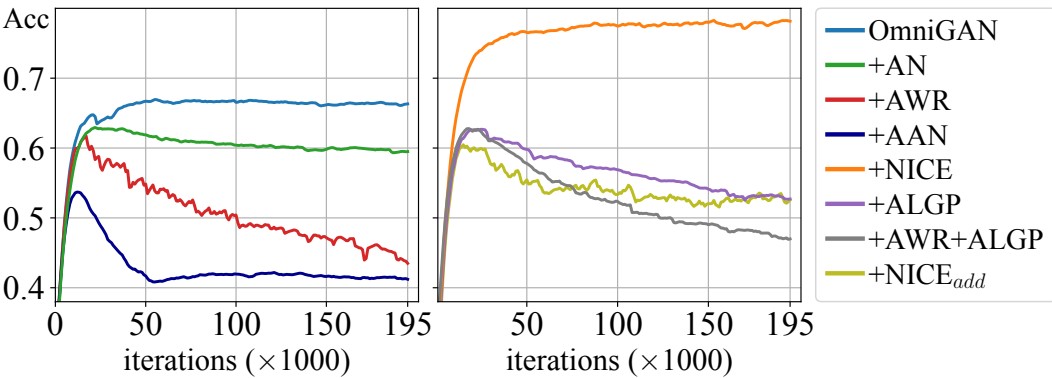

Figure 7: Classification accuracy of different methods. We input test images into the classifier of discriminator and assess its accuracy in correctly categorizing the images into their true classes (out of 10 categories in CIFAR-10). All other methods have lower accuracy than the baseline OmniGAN, but NICE preserves the classifier accuracy.

# G  Rationale behind choosing NICE over the direct gradient regularization

Consider the regularization term in Theorem 2 and its Taylor expansion. Indeed, this expression penalizes the squared norms of first- and second-order gradients of $f(\boldsymbol{a})$. However, notice that our expression has specific reweighting effect for penalizing the first- and second-order gradients. The specific gradient penalties

$$2\beta^2 \sum_k a_k^2 \nabla_k^2 f(\boldsymbol{a}) + \beta^4 \sum_{j,k} a_j^2 a_k^2 (H_{jk}^{(f)}(\boldsymbol{a}))^2 \tag{40}$$

emerge only in case of using the multiplicative noise drawn from $\mathcal{N}(\boldsymbol{1}, \beta^2 \boldsymbol{I})$ for Theorem 2. Importantly, the multiplicative noise $\boldsymbol{z} \sim \mathcal{N}(\boldsymbol{1}, \beta^2 \boldsymbol{I})$ does not introduce new semantics into feature vector $\boldsymbol{a}$ by operation $\boldsymbol{z} \odot \boldsymbol{a}$, *i.e.*, only feature semantics that are present in feature vector (features of $\boldsymbol{a}$ that are non-zero) that describe object are modulated. Yet, this multiplicative noise does help control the Rademacher Complexity (RC) due to Lemma 2 and Theorem 1. However, imposing a generic penalty of form $\beta(\sum_k \nabla_k^2 f(\boldsymbol{a}) + \sum_{j,k} (H_{jk}^{(f)}(\boldsymbol{a}))^2$ that arises when the additive noise $\boldsymbol{z} \sim \mathcal{N}(\boldsymbol{0}, \beta^2 \boldsymbol{I})$ is applied to $\boldsymbol{a}^{(i)}$, *i.e.*, $\boldsymbol{z} + \boldsymbol{a}$.

Let $\boldsymbol{z} \sim \mathcal{N}(\boldsymbol{0}, \beta^2 \boldsymbol{I})$. The consistency loss regularization with the additive noise injection is:

$$\mathbb{E}_{\boldsymbol{z}_1, \boldsymbol{z}_2}\big[\big(f(\boldsymbol{z}_1 + \boldsymbol{a}) - f(\boldsymbol{z}_2 + \boldsymbol{a})\big)^2\big] = 2\beta^2 \sum_k \nabla_k^2 f(\boldsymbol{a}) + \beta^4 \sum_j \sum_k (H_{jk}^{(f)}(\boldsymbol{a}))^2.$$

Such a variant means that semantics that are not present in $\boldsymbol{a}$ may be "activated" by the noise, drastically altering the meaning of $\boldsymbol{a}$ and damaging the information it carries about image/object.

While directly applying gradient penalization does not introduce noise, its connection to the additive noise suggests it will have a negative effect on semantics. Table 8 evaluates the multiplicative *vs.* additive noise modulators, and the direct penalties. Table 9 provides experimental results for different gradient regularization variants. Furthermore, Figure 7 shows that NICE obtains best accuracy than other variants, showing that the multiplicative noise modulation preserves the semantics better than other variants, justifying the rationale behind NICE.

Table 9: Results for ALGP (adaptive latent gradient penalization), NICE$_{add}$ (consistency regularization with additive noise), AWR (adaptive weight regularization) on 10% CIFAR-10/100 using OmniGAN ($d' = 256$).

| Method | 10% CIFAR-10 | | | 10% CIFAR-100 | | |
|---|---|---|---|---|---|---|
| | IS ↑ | tFID ↓ | vFID ↓ | IS ↑ | tFID ↓ | vFID ↓ |
| OmniGAN | 8.49 | 22.24 | 26.33 | 8.19 | 45.41 | 50.33 |
| +ALGP | 8.52 | 19.15 | 22.72 | 9.18 | 32.98 | 37.51 |
| +AWR+ALGP | 8.72 | 16.82 | 20.45 | 10.14 | 26.44 | 30.23 |
| +NICE$_{add}$ | 8.64 | 17.94 | 21.59 | 9.34 | 28.59 | 33.02 |
| +NICE | **9.26** | **7.23** | **11.08** | **11.50** | **16.91** | **21.56** |

# H  Training overhead with and without NICE

We provide number of parameters/multiply–accumulate (MACs) (for both generator and discriminator), number of GPUs and cost of time (seconds per 1000 images, secs/$k$img) in Table 10. With the efficient implementation (Figure 1), our NICE only introduces small fraction of cost of time on high-resolution datasets. 15.08% on ImageNet and 18.02% on FFHQ/low-shot datasets. Figure 8 illustrates the FID gain *vs.* speed, which shows that a slightly increase in training time yields substantial improvements, particularly when compared to recent state-of-the-art approaches like FakeCLR [32] and InsGen [63]. We believe the increased computational overhead is outweighed by the considerable benefits and the improvements justify the extra fraction of time.

Table 10: Number of parameters, MACs and secs/$k$img for models with *vs*. without the NICE. Experiments were performed on NVIDA A100 GPUs.

| Dataset | Archtecture | GPUs | Baseline | | | +NICE | | |
|---|---|---|---|---|---|---|---|---|
| | | | #Par. | MACs | sec/$k$img | #Par. | MACs | sec/$k$img |
| CIFAR-10 | OmniGAN ($d'$=256) | 1 | 8.51M | 2.79G | 0.80 | 8.51M | 3.87G | 1.15 |
| CIFAR-100 | OmniGAN ($d'$=256) | 1 | 8.81M | 2.79G | 0.81 | 8.81M | 3.87G | 1.17 |
| ImageNet | BigGAN | 2 | 115.69M | 18.84G | 1.79 | 115.69M | 20.43G | 2.06 |
| Low-shot & FFHQ | StyleGAN2 | 2 | 48.77M | 44.15G | 5.66 | 48.77M | 52.00G | 6.68 |

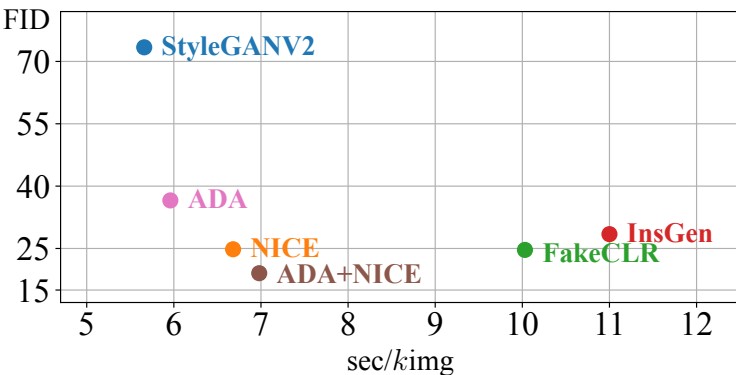

Figure 8: FID *vs*. sec/$k$img of different methods on 5 low-shot datasets. FID are averaged over the 5 low-shot datasets.

## I Qualitative comparison among DA, ADA, AACR and NICE+ADA

Figure 9 provides qualitative comparison between augmentation based method DA, ADA, AACR and NICE+ADA. We find DA, ADA and AACR leak augmentation clues to the generated images while NICE+ADA prevents the leakage. Figure 10 further demonstrates augmentation-based method, although can prevent discriminator overfitting problem by expanding data space through diverse augmentations, has the potential to cause increased gradient norm due to Prop. 1 and the fact ADA involves random additive/multiplicative noise. However, when combining NICE with ADA, the gradient issues caused by ADA were alleviated.

## J Generated images

Figures 11, 12, 13, 14 and 15 provide the generated images on CIFAR-10, ImageNet, CIFAR-100, low-shot datasets and FFHQ with or without NICE. We can see NICE improve the image generation quality.

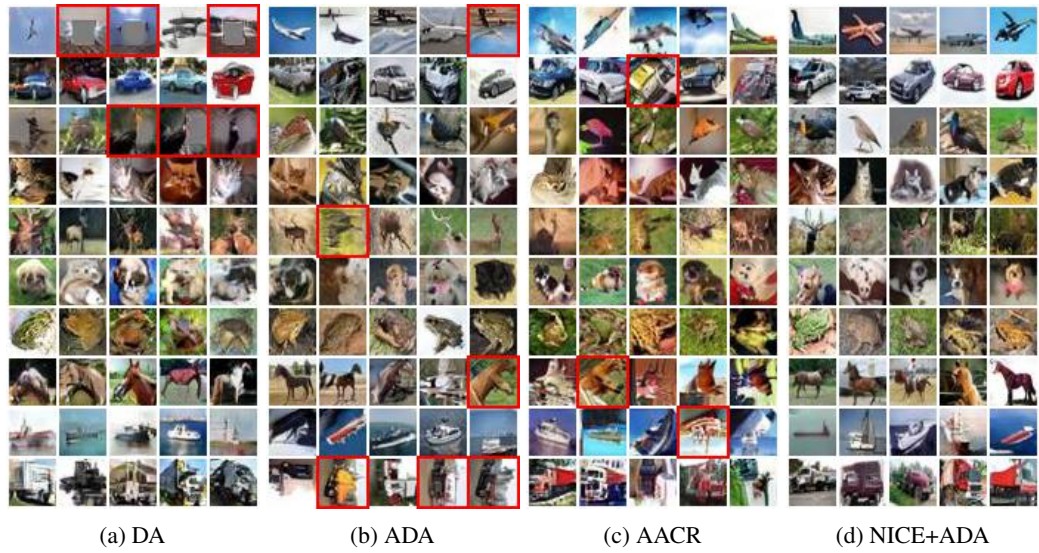

|                |                |                 |                    |
| :------------: | :------------: | :-------------: | :----------------: |
| (a) DA         | (b) ADA        | (c) AACR        | (d) NICE+ADA       |

Figure 9: Generated images using (a) DA (differentiable augmentation [68]), (b) ADA (adaptive differentiable augmentation [22]), (c) AACR (adaptive augmentation-based consistency regularization) and (d) NICE+ADA on 10% CIFAR-10 using OmniGAN ($d'$=256) [71] We use red boxes to bound the images that leak the augmentation. DA leaks the cutout augmentation. ADA and AACR leaks the rotation augmentation. Combining NICE with ADA prevents the leakage of augmentation.

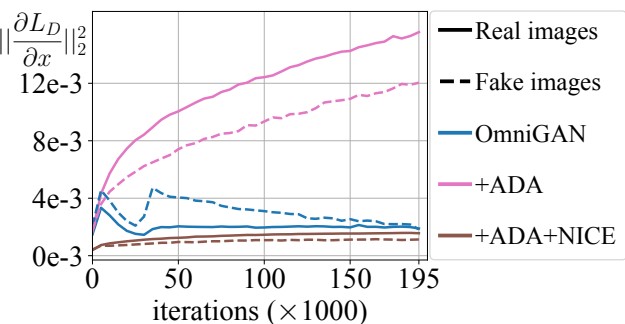

Figure 10: The gradient norm $\|\frac{\partial L_D}{\partial \boldsymbol{x}}\|_2^2$ of ADA and ADA+NICE.

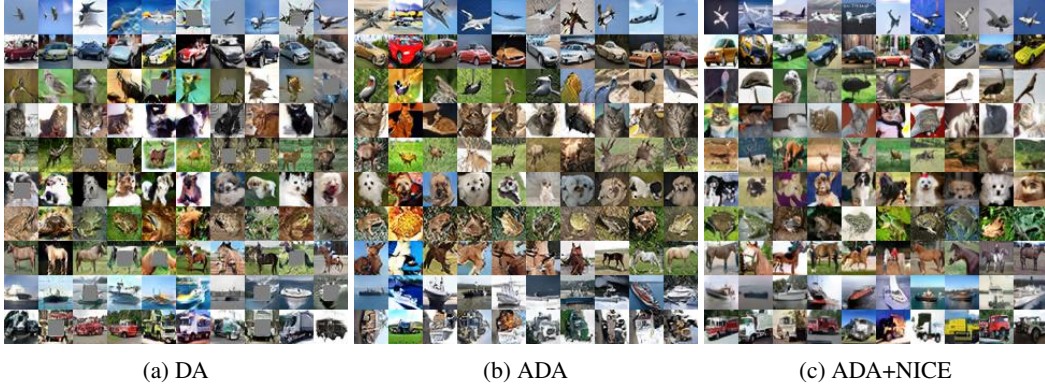

|                      |                      |                      |
| :------------------: | :------------------: | :------------------: |
| (a) DA               | (b) ADA              | (c) ADA+NICE         |

Figure 11: Generated images using (a) DA, (b) ADA and (c) NICE+ADA on 10% CIFAR-10 using OmniGAN ($d'$=1024). Note that DA leaks cutout augmentation and ADA leaks the rotation augmentation.

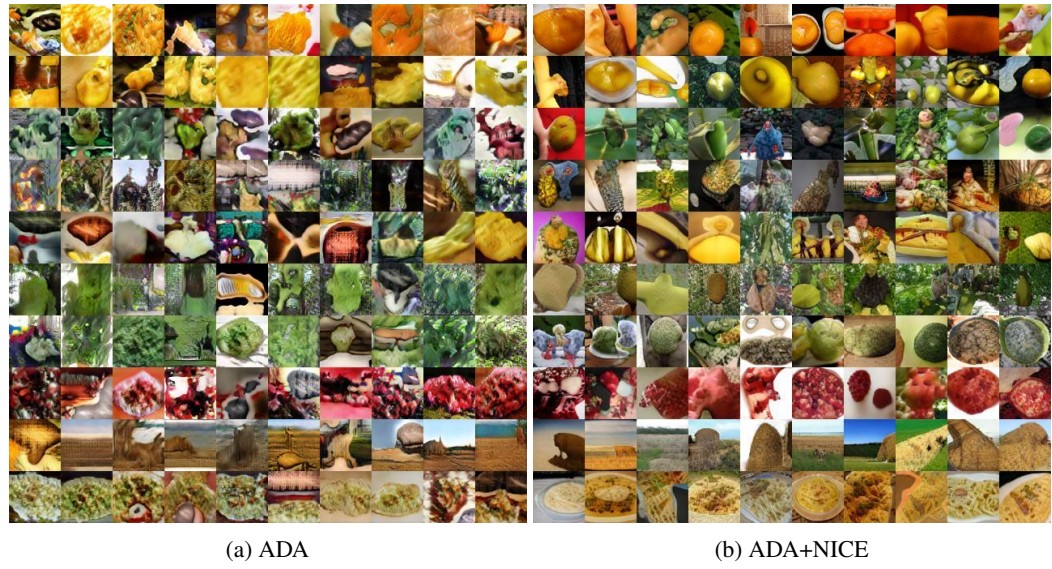

(a) ADA

(b) ADA+NICE

Figure 12: Generated images using (a) ADA and (c) ADA+NICE on 2.5% ImageNet using BigGAN. ADA struggles to capture the semantics of images, whereas NICE provides better image quality.

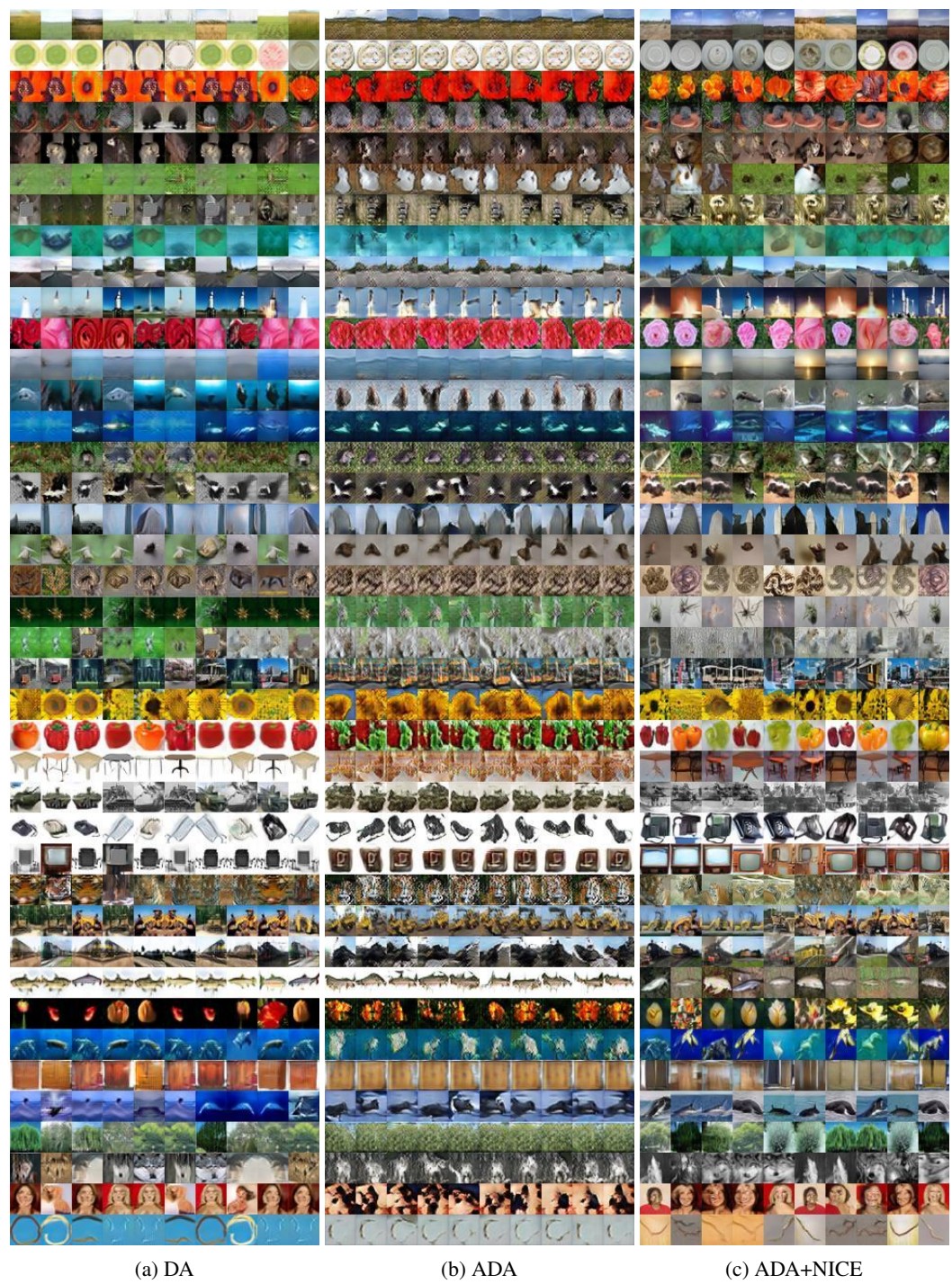

(a) DA (b) ADA (c) ADA+NICE

Figure 13: Generated images using (a) DA, (b) ADA and (c) NICE+ADA on 10% CIFAR-100 using OmniGAN ($d'$=1024). We present the last 40 classes out of the 100 classes 40. Note that DA leaks cutout augmentation clues (row 7 and row 21) and ADA generates images lacking diversity, while NICE+ADA provides better visual quality.

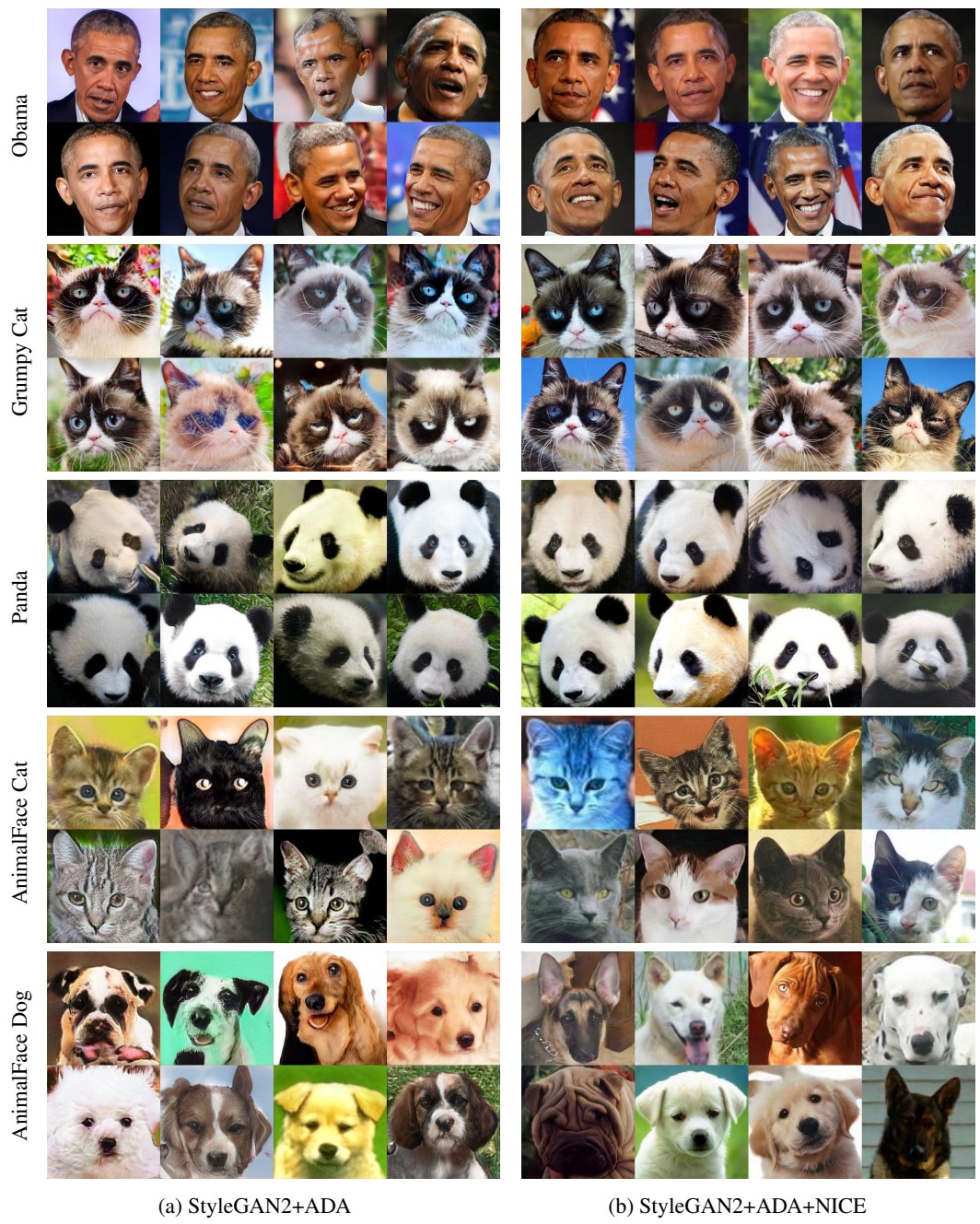

(a) StyleGAN2+ADA          (b) StyleGAN2+ADA+NICE

Figure 14: Qualitative comparison between ADA [22] and ADA+NICE on 100-shot and AnimalFace datasets. Adding NICE clearly improves the image quality.

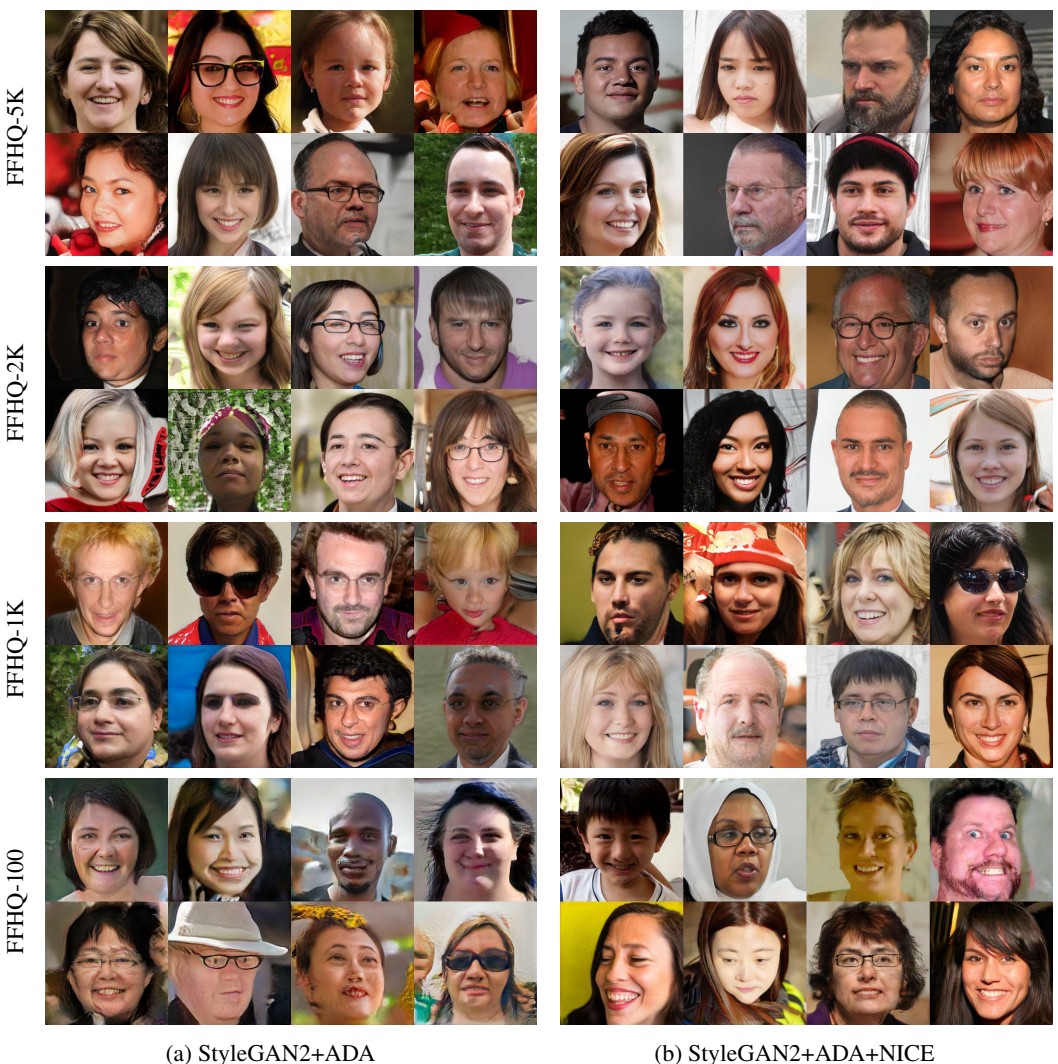

(a) StyleGAN2+ADA              (b) StyleGAN2+ADA+NICE

Figure 15: Qualitative comparison between ADA [22] and ADA+NICE on FFHQ dataset. Images are generated without trunction [25]. Adding NICE can provide better visual quality.

