# OpenReview forum: "NICE: NoIse-modulated Consistency rEgularization for Data-Efficient GANs"
_NeurIPS.cc/2023/Conference — NeurIPS 2023 poster_

### Official Review · Reviewer_ijuK · 2023-07-02

**Soundness:** 4 excellent
**Presentation:** 3 good
**Contribution:** 3 good
**Rating:** 8
**Confidence:** 2

**Summary:**

The authors propose a noise modulation and regularization scheme for GANs that reduces disciminator overfitting and improves training stability in the low-data scheme. The technique demonstrate consistent improvements when applied to several different network architectures and datasets.

**Strengths:**

The paper contains very extensive comparisons to competing previous methods.
The theoretical motivation for the proposed technique is very extensive (but also very hard for non-experts to understand).
The theoretical analysis sheds some light on the effectiveness of consistency regularization techniques that have shown promise previously.
The numerical results are excellent.
The paper is very math-heavy, but the authors do a good job of including some intuition of the different propositions and lemmas (e.g. lines 142-147, 156-160, 173-175, 191-196).
The inclusion of separate tFIDs and vFIDs is refreshing to see.

**Weaknesses:**

Not a single generated image is shown in the main paper. Given that some concerns about the reliability of FID have been raised as of late, visual results (e.g. the 100-shot results from the supplemental, Fig. 11) would be appreciated.

The paper mentions many similar-sounding terms (generation gap, generalization gap, output gap, generalization error, discriminator discrepancy) that might confuse non-experts. A brief description of the effect and its implications on training should be included for non-expert readers.

Many images in the supplemental are of unusably low quality. This is especially true for CIFAR results, which need to be integer-upscaled to preserve sharpness.

**Questions:**

I do not understand where the second derivatives come from in equation 9.
Equation 10 seems to perform a Taylor expansion, is this accurate? If so, the fact should be mentioned for clarity.

Is equation 1 a contribution of this paper or something presented in previous work?

What does MA in tables 1-4 mean?

In figure 2, the y-axis labels are incorrectly formatted (1e^2 instead of 1*10^2 or 1e2)

The authors claim (line 171) that proposition 1 can be extended to more complicated architectures - is an example of this shown somewhere, such as previous work?

The datasets used in Figures 2 and 3 don't seem to be mentioned

NICE seems to work best when combined with geometric augmentation techniques, especially ADA. A comment on this would be useful - ADA already minimizes the gap between the real and fake distributions, so what does NICE add in this case?

**Limitations:**

The limitations are not properly discussed in the main text. Is there a wall clock cost to the proposed method? Does it require changes to training hyper parameters (learning rate etc.)?

---

> ### Author Rebuttal · Authors · 2023-08-06
>
> # Response to Rev. 5 (ijuK)
>
> ***We thank the reviewer** for the constructive review and valuable questions that have helped us improve our work.*
>
> ## 1. Not a single generated image shown in the main paper.
>
> We apologize. We have selected now **images on 100-shot and AnimalFace
> datasets from Figure 11 (supplementary material)** and added them into the main draft.
>
>
> ## 2. The paper mentions many similar-sounding terms.
> Thank you for bringing our attention to multiple terms requiring disambiguation, which is invaluable in improving the clarity of our work and helping readers. The table below summarizes each the meaning of each term (we have added it now to our revised work):
> |Term|Meaning|
> |-|-|
> |generalization gap|the difference in a model's performance between the training data and unseen testing data|
> |output gap|the difference between the average discriminator output for real data and that for fake data|
> |discriminator discrepancy|the difference in the average output of the discriminator between two distributions|
> |generalization error|the error of the discriminator's predictions on unseen testing data|
>
> Additionally, we have duly rectified the term from "generation gap" to "generalization gap".
>
> ## 3. Many images in the supplemental are of unusably low quality.
>
> We apologize. We have now upscaled them accordingly given unlimited supplementary material size.
>
> ## 4. Where the second derivatives come from in Eq. 9?
>
> Thank you. They come from expanding $h(\cdot)$ around ${\bf a}\_f$ with the Taylor expansion. This sort of analysis and first- and second-order expansion are a typical starting point in several works:
> >* *What Regularized Auto-Encoders Learn from the Data-Generating Distribution?*, Alain & Bengio, JMLR'14
> >* *Sharpness-Aware Minimization for Efficiently Improving Generalization*, Foret et al, ICLR'21
> >* *How Does Mixup Help With Robustness and Generalization?* Zhang et al, ICLR'21
>
> As higher-order terms usually decay according $\mathcal{O}(\frac{1}{o!})$ where $o$ indicates the order, they are negligible.
>
> ## 5. Is Eq. 1 a contribution of this paper?
>
> Eq. 1 is a standard formulation for analyzing the generalization of GANs, as in:
> > *On the Discrimination-Generalization Tradeoff in GANs*, ICLR'18, Zhang et al.
>
> **We have updated it now with the equation given in Resp. 1 to Rev. 3 (LrCf)**, which is our contribution. We have also included extension of Fig. 5 and explanations from Sections B.2 and B.3 of the supplementary material to complement Fig. 1 in the main paper.
>
>
> ## 6. What does MA in Tables 1-4 mean?
>
> We apologize. MA denotes `massive augmentations' (including DA and ADA) first used in:
> > *Generative co-training for generative adversarial networks*, AAAI'22, Cui et al.
>
> DA and ADA are:
> > *Differentiable augmentation for data-efficient GAN training*, NeurIPS'20, Zhao et al.
> \
> *Training generative adversarial networks with limited data*, NeurIPS'20, Karras et al.
>
> ## 7. Claim that Prop. 1 can be extended to more complicated architectures.
>
> We believe this is a misunderstanding (we revised now the language). Our aim was to emphasize that the beneficial effect of implicit weight regularization achieved through noise modulation, despite being analyzed within a simplified two-layer system, remains pertinent for networks encompassing multiple layers, including convolutional neural networks (indeed, we use NICE across several layers). This is attributed to the fact that networks with multiple layers can be conceptually treated as an aggregation of two layers, and a convolutional layer is a specialized case of a linear layer.
>
>
> These propositions find validation in the following references:
> >* *On Dropout and Nuclear Norm Regularization*, Mianjy et al, ICML'19
> >* *The Implicit and Explicit Regularization Effects of Dropout*, Wei et al, ICML'20
> >* *Dropout: Explicit Forms and Capacity Control*, Arora, ICML'21
>
>
> ## 8. Datasets used in Figures 2 and 3 do not seem to be mentioned.
>
> We apologize. In Fig. 2 and 3, we use a 10% data, CIFAR-10, OmniGAN ($d'=256$).
>
> ## 9. ADA already minimizes the gap between the real and fake distributions, so what does NICE add in this case?
> Thank you for the interesting question. Below we analyze ADA's underlying issues. While both ADA and NICE effectively prevent overfitting, they employ differing strategies. NICE reduces the Rademacher complexity of the discriminator while ADA expands the data space through diverse augmentations. However, **NICE possesses an additional advantage which ADA lacks. It implicitly penalizes gradients** (e.g., minimizing Eq. 10 implies minimizing gradient norms $f'^2(\cdot)$ and $f''^2(\cdot)$),  thereby improving the stability of GAN training.
> \
> \
> In contrast, ADA lacks such properties. With augmentation techniques that involve random noise additions or multiplications, an analysis based on Eq. 9 suggests that ADA might in fact lead to an increase in gradient norms. Empirical observations support this supposition: we encountered increase in gradient norms for ADA (**kindly see Figure 4 in the rebuttal PDF**). However, when combining NICE with ADA, the gradient issues caused by ADA were alleviated, giving the best of two worlds (**ADA and NICE are complementary**).
>
> ## 10. Limitations, training cost and whether it requires changes to training hyper-parameters (learning rate et.?)
> We have now moved the limitations addressed in Section I of the supplementary materials to the main paper. While it is true that implementing NICE comes with small added training costs, our design (see Fig. 5, supplementary materials) enables efficient parallelization of the process. Moreover, even without parallelization, the incremental cost compared to the baseline is minimal, as illustrated in Figure 5 in the rebuttal PDF and Table 7 in the supplementary materials. NICE is built on GAN backbone architectures, and we keep all the learning hyper-parameters of original backbones untouched.

---

> > ### Comment · Reviewer_ijuK · 2023-08-13
> >
> > Thank you for the thorough response. My concerns have been adequately addressed.
> >
> > I will follow the other discussions here and reconsider my rating if need be.

---

> > > ### Author Response · Authors · 2023-08-15
> > >
> > > Thank you for your kind and constructive response.
> > >
> > > Authors

---

### Official Review · Reviewer_m7vf · 2023-07-06

**Soundness:** 3 good
**Presentation:** 2 fair
**Contribution:** 2 fair
**Rating:** 6
**Confidence:** 4

**Summary:**

This paper proposed NICE, a technique that enforces the discriminator to be consistent with the same inputs under different noise modulations. The authors showed us both in theory and practice that NICE is effective at preventing discriminator overfitting and achieves superior performance in image generation under limited data settings.

**Strengths:**

1. The method is based on theoretical proof, and the experimental results show the effectiveness.

2. The method achieves competitive performance against many existing methods. The evaluation involves multiple GAN models, and baseline methods.

**Weaknesses:**

1. The authors show the computational overhead of NICE in appendix, which is not small and it could increase the training time by a large amount. There is no quantitative measure for scalability of NICE. For example, what will happen when the image resolution is 1080p or even higher, or when the dataset is quite large. What is the training cost?

2. NICE introduces some new hyperparameters but it is not clear how they are tuned. What's their sensitivity to different GAN models and data distribution (or datasets)? What's their sensitivity to baselines in Table 3?

**Questions:**

Please refer to “Weaknesses".

It seems that authors missed a strong baseline method SSGAN-LA [1] in Table 3 which could achieved a better performance. It is better to compare with missing baselines.

[1] Self-Supervised GANs with Label Augmentation, Neurips 2021.

**Limitations:**

Please refer to “Weaknesses" for limitations. The authors adequately addressed the potential negative societal impact of their work.

---

> ### Author Rebuttal · Authors · 2023-08-06
>
> # Response to Rev. 4 (m7vf)
>
> ***We thank the reviewer** for the constructive review and valuable questions that have helped us improve our work.*
>
> ## 1. The computational overhead of NICE.
>
> Kindly notice **our computational overhead is small**. Firstly, our usage of multiplicative noise modulation as in **Fig. 5 (supplementary material)** and Fig. 1 (main paper), **is restricted to the last 3 or 4 blocks**, i.e.,  $L-l+1$, of the discriminator with $1,\cdots,L$ blocks. Moreover, **these last few blocks are responsible for handling low resolution feature maps and can be readily parallelized**. Even without parralelization, increase is by 0.35sec per 1000 images on OmniGAN. On StyleGAN2, which is a slower network, the extra time required by NICE was only 18\% on top of StyleGAN2. Meanwhile, **Fig. 5 in the rebuttal pdf illustrates the FID gain *vs.* speed**, which shows that **a slightly increase in training time yields substantial improvements**, particularly when compared to recent state-of-the-art approaches like FakeCLR and InsGen. We believe the increased computational overhead is outweighed by the considerable benefits and the improvements justify the extra fraction of time.
>
> ## 2. What will happen when the image resolution is 1080p or even higher?
>
> **As our module is used only  for the last 3 or 4 blocks of the discriminator**, i.e.,  $L-l+1$, this means they are not exposed to high resolution feature maps. (see Fig. 5 (supplementary material)). **Table below** shows no large speed increase in such a case.
>
> Below we apply NICE module to the StyleGAN2 for the last 4 blocks, and provide the sec/$k$img given resolutions as follows:
> |Resolution|$512\times512$|$1024\times1024$|$2048\times2048$|
> |-|:-:|:-:|:-:|
> |StyleGAN2|18.65|51.76|143.25|
> |StyleGAN2+NICE|20.74|56.03|147.76|
> |relative increase|11.17\%|8.25\%|**3.05\%**|
>
> As our NICE is implemented on last few blocks of the discriminator, which are low resolution feature maps, **the relative computation cost increase is actually smaller as the remaining standard network layers dominate compute time**.
>
>
> ## 4. What hyper-parameters do you use and what is their sensitivity to different GAN models?
> We keep the learning hyperparameters of the original GAN backbones untouched. NICE introduces four new hyperparameters: the place to apply NICE within a block $c\in\\{C_1, C_2, C_R\\}$, which blocks to use $l\in\\{1,...L\\}$, the threshold $\eta$ and the regularization strength $\Delta_\gamma$. $c$ can be simply choosed from $C_1C_2C_R$ and $C_1C_2$ (we use $C_1C_2C_R$ for OmniGAN and BigGAN, and $C_1C_2$ for StyleGAN2). $l$ is often applied for the last 3 or 4 blocks. $\eta$ is fixed as 0.5 for BigGAN and OmniGAN. On StyleGAN2, we recommand lower $\eta$ for diverse datasets. we set $\eta=0.6$ for FFHQ dataset $\eta=0.9$ for the 5 low-shot dataset. Generally ($c=\\{C_1, C_2\\}$, $l=\\{L-3, ..., L\\}$, $\eta=0.6$, $\Delta_\gamma=0.05$) is a good starting point when applied to new dataset.
>
> Below, we varied the hyper-parameters on the 5 low-shot dataset (Obama, Grumpy Cat, Panda, AnimalFace Cat, Animal Face Dog). **A slight change the hyper-parameters does not make a big difference regarding the results**, showing that our NICE is not too sensitive to hyper-parameters:
>
> |$l\in\\{1...L\\}$|Obama|Grumpy Cat|Panda|AnimalFace Cat|AnimalFace Dog|
> |-|:-:|:-:|:-:|:-:|:-:|
> |4,5,6|25.66|20.01|9.25|26.15|48.18|
> |3,4,5,6|24.56|18.78|8.92|25.25|46.56|
> |2,3,4,5,6|26.35|19.34|9.14|25.61|47.68|
>
> |$c\in\\{C_1, C_2, C_R\\}$|Obama|Grumpy Cat|Panda|AnimalFace Cat|AnimalFace Dog|
> |-|:-:|:-:|:-:|:-:|:-:|
> |$C_1$|25.77|19.42|8.98|24.97|47.17|
> |$C_1,C_2$|24.56|18.78|8.92|25.25|46.56|
> |$C_1,C_2,C_R$|26.51|19.51|9.56|26.18|46.93|
>
> |$\Delta_\gamma$|Obama|Grumpy Cat|Panda|AnimalFace Cat|AnimalFace Dog|
> |-|:-:|:-:|:-:|:-:|:-:|
> |0.01|26.48|19.1|9.09|25.33|47.15|
> |0.05|24.56|18.78|8.92|25.25|46.56|
> |0.1|25.46|19.52|9.01|25.42|47.06|
> |0.2|25.74|19.85|9.13|25.30|47.87|
>
> |$\eta$|Obama|Grumpy Cat|Panda|AnimalFace Cat|AnimalFace Dog|
> |-|:-:|:-:|:-:|:-:|:-:|
> |0.95|25.20|20.06|9.07|25.20|46.44|
> |0.90|24.56|18.78|8.92|25.25|46.56|
> |0.80|25.71|18.90|9.15|26.05|47.43|
>
> ## 5. Authors missed a strong baseline method SSGAN-LA [1] in Table 3.
>
> SSGAN-LA [1] did not perform experiments on the five low-shot datasets, 100-shot Obama/Grumpy cat/Panda
> and Animal Face Cat/Dog datasets. They only released code for BigGAN.
> \
> \
> Thus, we have followed the authors' code implementation for BigGAN and re-implemented it for StyleGAN2 on the five low-shot datasets. For SSGAN-LA, we tried the multi-hinge loss and the cross_entropy loss provided in the code of SSGAN-LA. We obtained the best the results with the multi-hinge loss for StyleGAN2+SSGAN-LA but our StyleGAN2+NICE is still a stronger performer:
> |Dataset|Obama|Grumpy Cat|Panda|AnimalFace Cat|AnimalFace Dog|
> |-|-|-|-|-|-|
> |StyleGAN2+SSGAN-LA|79.88|38.42|28.6|78.78|109.91|
> |StyleGAN2+NICE (ours)|**24.56**|**18.78**|**8.92**|**25.25**|**46.56**|

---

> > ### Comment · Reviewer_m7vf · 2023-08-20
> >
> > I have read the authors' response and I find that the current state of this paper is satisfying, though some aspects of this paper could be improved. I have raised my rating. Thank the authors for their effort during the discussion phases.

---

> > > ### Author Response · Authors · 2023-08-20
> > >
> > > Thank you. We really appreciate the reviewer's comments we received and help in shaping up our work.

---

### Official Review · Reviewer_LrCf · 2023-07-06

**Soundness:** 3 good
**Presentation:** 3 good
**Contribution:** 2 fair
**Rating:** 5
**Confidence:** 4

**Summary:**

This paper proposes a training approach called NoIse-modulated Consistency rEgularization (NICE) to improve the data-efficiency of generative adversarial networks (GANs) by addressing issues related to limited data. It introduces adaptive multiplicative noise into the discriminator to modulate its latent features, preventing discriminator overfitting. To mitigate the instability of GAN training caused by increased gradient norm, a constraint is imposed on the discriminator to ensure consistency for the same inputs under different noise modulations. The experimental results demonstrate the effectiveness of NICE in reducing discriminator overfitting and improving the stability of GAN training, achieving state-of-the-art results on various datasets and low-shot generation tasks.

**Strengths:**

- The paper is well-organized and well-written, making it easy to understand its contribution.
- The paper provides theoretical analysis to understand the connection between introducing multiplicative Gaussian noise to the discriminator and GAN generalization. The authors also uncover the negative impact of simple noise multiplication on gradient norm and propose a noise-modulated consistency regularization with theoretical grounds to improve it.
- Experimental results presented in the paper validate the effectiveness of the proposed method, surpassing the baseline in most experiments.

**Weaknesses:**

- The complete objective function of the proposed method should be included in the main text, rather than in the appendix.
- In the GAN field, it is suggested to conduct multiple experiments and report the results in terms of mean and standard deviation due to the random variation of the results caused by different trials.
- According to the theoretical analysis in this paper, it is easy to understand the motivation behind the introduction of Eq12, but the motivation behind Eq13 and Eq14 is unclear. Although the ablation experiments in Table 6 empirically demonstrated the benefits of the proposed method, I still expect the authors to provide a more explicit explanation for these equations. Moreover, it is recommended to report the experimental results using only Eq12 and Eq14, as this would further demonstrate the effectiveness of the method.

**Questions:**

- Does Eq1 encompass all GAN objective functions, such as the original min-max and non-saturating GANs? It is suggested that the authors can provide detailed explanations, as the theoretical work of this paper is based on Eq1.

Some typos:
- Lack parentheses in Eq9.
- f(x) -> f(\alpha) in Line 185.
- N(0,\beta^2 I^d) -> N(1,\beta^2 I^d) in Line 200.

**Limitations:**

No, the authors have not addressed the limitations and potential negative societal impact of their work.

---

> ### Author Rebuttal · Authors · 2023-08-06
>
> # Response to Rev. 3 (LrCf)
>
> ***We thank the reviewer** for the constructive review and valuable questions that have helped us improve our work.*
>
> ## 1. The complete objective function of the proposed method should be included in the main text, rather than in the appendix.
>
> Thank you. Absolutely. We have now combined details from Sections B.2 and B.3 of the supplementary material. In general, we have improved the notation and Eq. 12-14 can be directly injected into the GAN objective by $\Omega(\cdot)$. To that end, we have compacted the objective as follows:
>
> $
> \begin{cases}
> L_D^\text{AN}=\min\limits\_{{\boldsymbol\theta}_d}\mathbb{E}\_{\bf x\sim\nu_n}\[h\_\text{AN}({\bf x}; {\boldsymbol\theta}_d)\\!+\\!\gamma\Omega({\bf x})\]+\mathbb{E}\_{\bf x\sim\hat{\mu}_m}\[ -h\_\text{AN}({\bf x}; {\boldsymbol\theta}_d)\\!+\\!\gamma\Omega({\bf x})\]\\\\
> L_G^\text{AN}=\min\limits\_{{\boldsymbol\theta}_g}\mathbb{E}\_{\bf z\sim p\_z}\[-h\_\text{AN}(g({\bf z}; {\boldsymbol\theta}_g))\\!+\\!\gamma\Omega(g({\bf z}; {\boldsymbol\theta}_g))\]
> \end{cases}\\;\text{where}\\;\\;\Omega({\bf x})=||f\_{1}({\bf x})-f\_{2}({\bf x})||_2^2,
> $
>
> where $h\_\text{AN}({\bf x}; {\boldsymbol\theta})$ denotes our modified discriminator inclusive of the noise injection,  ${\boldsymbol\theta}_d$ and  ${\boldsymbol\theta}_g$ are learnable parameters of discriminator and generator, and $f\_1(\cdot)$ and $f\_2(\cdot)$ are two feature extractors with two different multiplicative noise injections, as per Fig. 5 in the supplementary material. Moreover, $\nu_n$, $\hat{\mu}_m$ and $p\_z$ represent the finite fake distribution, the empirical training distribution and distribution of the generator input. $\gamma\geq 0$ is the regularization hyper-parameter.
>
> ## 2. Conduct multiple experiments and report the mean and standard deviation.
>
> Thank you. In fact, we have run 5 trials for NICE and NICE+ADA in Tables 1-5 and reported the mean. As the standard deviations were less than 1% relatively, we followed the practice of most GAN papers (listed below) which typically do not report the standard deviations:
>
>  >* *FakeCLR: Exploring Contrastive Learning for Solving Latent Discontinuity in Data-Efficient GANs*, ECCV'22, Li et al.
>  >* *DigGAN: Discriminator gradIent Gap Regularization for GAN Training with Limited Data*, NeurIPS'22, Fang et al.
>  >* *Differentiable Augmentation for Data-Efficient GAN Training*, NeurIPS'20, Zhao et al.
>
> However, **Tables 1-4 in the rebuttal PDF contain full results for ADA+NICE with STD** supplemented as per your request.
>
> ## 3. The motivation for Eq. 12 is clear but motivation of Eq. 13-14 is unclear.
>
> Thank you. This is a very interesting question.
> \
> \
> Eq. 12 provides regularization for discriminator given the real data. In the same spirit, the discriminator has to be regularized w.r.t. the fake data, as in Eq. 13, to stablizing GAN training. Regularizing both real and fake data was explored in several papers:
> >* *Which Training Methods for GANs do actually Converge?*, ICML'18, Mescheder et al.
> >* *Improving Generalization and Stability of Generative Adversarial Networks*, ICLR'19, Thanh-Tung et al.
> >* *Stabilizing Training of Generative Adversarial Networks through Regularization*, NeurIPS'17, Roth et al.
> >* *DigGAN: Discriminator gradIent Gap Regularization for GAN Training with Limited Data*, NeurIPS'22, Fang et al.
>
> These works have found that regularizing gradients on both real and fake samples improves convergence and stability of  GAN training. Thus, we also apply NICE on both real and fake samples. In terms of why we use NICE$\_{G_f}$, it can be explained as reducing first- and second-order gradients of discriminator while optimizing generator parameters ${\boldsymbol\theta}\_g$. Below we compare the Taylor expansions of generator objective without NICE$\_{G_f}$ *vs.* with NICE$\_{G_f}$ (in both cases NICE$\_{D_r}$ and NICE$\_{D_f}$ are switched on). Let ${\bf a}\_f$ be some fake sample from the generator. We optimize parameters ${\boldsymbol\theta}\_g$ of generator by minimizing:
>
> * without NICE$\_{G_f}$:
> $-h\_\text{AN}({\bf a}\_f)\approx-h({\bf a}\_f)-\frac{\beta^2}{2}\frac{\partial^2 h}{\partial f^2}|\_{{\bf a}_f}f''({\bf a}_f){\bf a}_f^2$
> * with NICE$\_{G_f}$:
> $-h\_\text{AN}({\bf a}\_f)+\gamma\Omega({\bf a}\_f)\approx-h({\bf a}\_f)-\frac{\beta^2}{2}\frac{\partial^2 h}{\partial f^2}|\_{{\bf a}_f}f''({\bf a}_f){\bf a}_f^2+\underbrace\{2\gamma\beta^2{\bf a}_f^2f'^2({\bf a}_f)+\gamma\beta^4{\bf a}_f^4f''^2({\bf a}_f)}\_{\text{grad. norms}}$,
>
> where $f'^2(\cdot)$ and $f''^2(\cdot)$ are simply squared norms of first- and second-order gradients (see underbrace in the eq.) of discriminator.
> \
> \
> This means that without NICE$\_{G_f}$, the generator is trained to generate images to the point where the gradient norms of discriminator can become large. In contrast, with NICE$\_{G_f}$, for some well-chosen $\gamma>0$, these first- and second-order gradient norms of discriminator are reduced by optimizing ${\boldsymbol\theta}\_g$ of generator, as they show up in the Taylor expansion.
> \
> \
> In that sense, Eq. 14 is also meaningful because while we are optimizing here the generator, the output of generator is still passed via discriminator. Thus, it makes sense to seek parameters of generator that not only improve the generator to ``fool'' discriminator but also stablizing the GAN training.
> \
> \
> To conclude, stabilizing GAN training can be achieved with Eq. 12-14. **See figure 2 in rebuttal PDF** for the gradient analysis when we ablate the Eq 12-14.
>
> ## 4. Report the experimental results using only Eq. 12 and Eq. 14.
> Kindly see response in the General Rebuttal (for all reviewers) due to the 6000 characters limited space.
>
> ## 5. Does Eq. 1 encompass all GAN objective functions, i.e., the original min-max and non-saturating GANs?
> Kindly see response in the General Rebuttal (for all reviewers) due to the limited space.

---

### Official Review · Reviewer_v2y4 · 2023-07-07

**Soundness:** 3 good
**Presentation:** 3 good
**Contribution:** 3 good
**Rating:** 7
**Confidence:** 4

**Summary:**

This paper proposes a regularization method called noise-modulated consistency regularization (NICE) to train GANs with limited data. In this method, this paper proposes modulating the discriminator's latent features using noise and imposing a constraint on the discriminator so that the middle outputs of the discriminator (particularly, the outputs before the prediction head) for differently modulated data are the same. This paper also provides a theoretical analysis, in which it is shown that the proposed regularization penalizes the first- and second-order gradients of latent features and improves the GAN training stability. The effectiveness of the proposed method was demonstrated using small-scale typically-used datasets, including CIFAR-10, CIFAR-100, ImageNet, and FFHQ datasets, and in low-shot generation tasks.

**Strengths:**

1. The effectiveness of the proposed method was demonstrated in various scenarios, including evaluation on small-scale typically-used datasets (CIFAR-10, CIFAR-100, ImageNet, and FFHQ datasets) and evaluation in low-shot generation tasks. In many scenarios, the proposed method achieves state-of-the-art performance while comparing with various baselines. Furthermore, the applicability of the proposed method was also demonstrated by applying the proposed method to various GANs (e.g., BigGAN and OmniGAN in Tables 2 and 3, and StyleGAN2 in Table 3) and using the proposed method with orthogonal methods (e.g., LeCam, DA, and ADA). Ablation studies are also conducted.

2. Not only the effectiveness of the proposed method is demonstrated, but also the theoretical analysis is provided. This analysis verifies the proposed method is useful for regularizing the first- and second-order gradients of latent features and improving the GAN training stability. This explanation is reasonable.

3. This paper is well written and easy to read. Although some explanation is slightly too concise, the discussion on related work is thorough.

**Weaknesses:**

1. Through theoretical analysis, I understand that the proposed method is useful for regularizing the first- and second-order gradients of latent features. However, this analysis raises a question of what happens when regularizing the first- and second-order gradients of latent features directly. As discussed in related work, there are several previous studies that propose gradient regularizations. I guess that the proposed method is better than a direct regularization method in terms of calculation cost; however, I would appreciate it if I could hear the opinion from the authors.

2. I cannot find the discussion on the increase in the calculation cost. I suspect the proposed method increases the calculation cost because the discriminator needs to process data twice, compared to a standard discriminator. For a fair comparison, it would be better to be discussed.

3. Some results are excluded in Tables 1–3 (e.g., IS/tFID for DigGAN in Table 1). I cannot find a clear explanation for why the results are excluded. It seems that DigGAN is one of the comparable baselines; therefore, I would appreciate it if the authors could provide the missing scores.

**Questions:**

1. What happens when directly regularizing the first- and second-order gradients of latent features? (See weakness 1)
2. Discuss the calculation cost. (See weakness 2)
3. Why are some results excluded in Tables 1–3. Provide the scores if possible (See weakness 3)

**Limitations:**

1. Image synthesis technologies could be misused for synthesizing fake images. In particular, few-shot image synthesis technologies will make it easy to synthesize such images because they reduce the data collection cost. It would be better to discuss the social impact in this aspect.
2. Although the versatility of the proposed method is demonstrated, I suspect that there may be some previous methods that are not compatible with the proposed method. Discussing this will be useful for future research.

---

> ### Author Rebuttal · Authors · 2023-08-05
>
> # Response to Rev. 2 (v2y4)
>
> ***We thank the reviewer** for the constructive review and valuable questions that have helped us improve our work.*
> ## 1. What happens when regularizing the first- and second-order gradients of latent features directly?
>
> Thank you. Consider the regularization term in Eq. 10 (main paper) and its Taylor expansion. Indeed, this expression penalizes the squared norms of first- and second-order gradients of $f({\bf a}_i)$. However, notice that our expression has specific penalty weights $2\beta^2||{\bf a}_i||_2^2$ and $\beta^4||{\bf a}_i||_2^4$ for penalty $||f'({\bf a}_i)||^2_2$ and  $||f''({\bf a}_i)||^2_2$ respectively. In fact, these specific penalty weights with the penalty given by
>
> $2\beta^2||{\bf a}_i||_2^2||f'({\bf a}_i)||^2_2 + \beta^4||{\bf a}_i||_2^4||f''({\bf a}_i)||^2_2$
> \
> emerge only in case of using multiplicative noise drawn from $\mathcal{N}({\bf 1},\beta^2{\bf I})$ for Eq. 9 and 10.
> \
> \
> **Importantly, see Resp. 4 to Rev. 1 (fQTZ)**. In that response we demonstrate that:
> * **the multiplicative noise ${\bf z}\sim\mathcal{N}({\bf 1},\beta^2{\bf I})$ does not introduce new semantics into feature vector ${\bf a}_i$ by operation ${\bf z}\cdot{\bf a}_i$**, i.e., only feature semantics that are present in feature vector (features of ${\bf a}_i$ that are non-zero) that describe object are modulated.
> * yet, **this multiplicative noise** does help control the Rademacher Complexity (RC) due to Lemma 2, Eq. 7 (main paper).
>
> We suspect that by ``regularizing the first- and second-order gradients of latent features directly'', Rev. 2 (v2y4) means imposing a generic penalty of form $\beta_1||f'({\bf a}_i)||^2_2 + \beta_2||f''({\bf a}_i)||^2_2$ or its variant  $2\beta^2||f'({\bf a}_i)||^2_2 + \beta^4||f''({\bf a}_i)||^2_2$ that  arises when the additive noise ${\bf z}\sim\mathcal{N}({\bf 0},\beta^2{\bf I})$ is applied to ${\bf a}_i$, i.e., ${\bf z}\cdot{\bf a}_i$.
>
> Let ${\bf z}\sim\mathcal{N}({\bf 0}, \beta^2{\bf I})$. we use Taylor expansion to expand the additive noise modulated consistency regularization as follows:
> $\mathbb{E}_{{\bf z}_1, {\bf z}_2}||f({\bf a}_i+{\bf z}_1)-f({\bf a}_i+{\bf z}_2)||^2\approx \mathbb{E} _{{\bf z}_1, {\bf z}_2}||({\bf z}_1-{\bf z}_2)f'({\bf a}_i)+\frac{{\bf z}_1^2-{\bf z}_2^2}{2}f''({\bf a}_i)||^2=2\beta^2||f'({\bf a}_i)||^2_2 + \beta^4||f''({\bf a}_i)||^2_2$
>
> **Such a variant means that semantics that are not present in ${\bf a}_i$ may be ``activated'' by the noise, drastically altering the meaning of ${\bf a}_i$ and damaging the information it carries about image/object.** (see  **Resp. 4 to Rev. 1 (fQTZ)**).
> \
> While directly applying gradient penalization with $\beta^2||f'({\bf a}_i)||^2_2 + \beta^4||f''({\bf a}_i)||^2_2$ does not introducing noise, It's connection to the additive noise suggests it will have a negative effect on semantics.
>
> Kindly **see the table in Resp. 4 to Rev. 1 (fQTZ) that evaluates multiplicative vs. additive noise modulators, and the direct penalties.**
>
> ALGP (adaptive latent gradient penalization), NICE$_{add}$ (consistency regularization with additive noise),  AWR (adaptive weight regularization)
> 10\% CIFAR-10/100 on OmniGAN ($d'=256$),
> |Dataset||10\% CIFAR-10|||10\% CIFAR-100||
> |-|-:|:-:|:-|-:|:-:|:-|
> ||IS|tFID|vFID|IS|tFID|vFID|
> |OmniGAN|8.49|22.24|26.33|8.19|45.41|50.33|
> |+ALGP|8.52|19.15|22.72|9.18|32.98|37.51|
> |+AWR+ALGP|8.72|16.82|20.45|10.14|26.44|30.23|
> |+NICE$_{add}$|8.64|17.94|21.59|9.34|28.59|33.02|
> |+NICE|**9.26**|**7.23**|**11.08**|**11.50**|**16.91**|**21.56**|
>
> Kindly see **Figure 3 in the rebuttal pdf**, which plots the classification accuracy of the discriminator on testing images for different variants. AN achieves higher accuracy than AAN (adaptive additive noise) and AWR. NICE obtains best accuracy than other variants, showing that the multiplicative noise modulation preserves the semantics better than other variants. This answers why we not directly penalize the gradient norms or directly regularize the weight norms, as they drastically alter the semantics of the features.
>
> ## 2. Is the proposed method better than a direct regularization in terms of calculation cost? What is the increase in the calculation cost?
>
> The specific penalty weights are important, as elaborated above. The computations are also simpler as we do not have to directly tap into the backpropagation and gradients to penalize them.
>
> Kindly notice that only last few blocks of discriminator (blocks $l,\cdots,L$) use our noise modulator penalty (kindly see Fig. 5 in the supplementary material). This design is also simple and very easy to parallelize. Even without parallelization, **the extra cost compared with the baseline is negligible**, as shown in **Figure 5** in the rebuttal pdf  and in **Table 7** of supplementary material.
>
>
> ## 3. Some results are excluded in Tables 1–3. DigGAN is one of the comparable baselines.
>
> * We directly use the code provided by the DigGAN and test the IS, tFID, and vFID, and obtain the results for BigGAN+DigGAN ($d'=256$) on CIFAR10 and CIFAR100 as follows:
> |Dataset||100\%|||20\%|||10\%||
> |-|-|-|-|-|-|-|-|-|-|
> ||IS|tFID|vFID|IS|tFID|vFID|IS|tFID|vFID|
> |CIFAR-10 BigGAN+DigGAN|9.28|5.33|9.35|8.81|13.28|17.25|8.32|18.54|22.45|
> |CIFAR-10 BigGAN+NICE|**9.50**|**4.19**|**8.24**|**8.96**|**8.51**|**12.54**|**8.73**|**13.65**|**17.75**|
> |CIFAR-100 BigGAN+DigGAN|**11.15**|8.13|13.06|9.98|16.87|21.59|**9.04**|23.10|27.78|
> |CIFAR-100 BigGAN+NICE|10.99|**6.13**|**11.08**|**10.32**|**13.17**|**17.80**|8.96|**19.53**|**24.33**|
>
> * As the authors did not release the code for StyleGAN2 on the five low-shot datasets, we reproduce their method and swap the hyperparameter ($\lambda\in\\{10, 20, 50, 100, 150\\}$ in their paper) for this tasks. We obtain the best results with $\lambda=20$ for (DigGAN+ADA) as follows:
> |Dataset|Obama|Grumpy Cat|Panda|AnimalFace Cat|AnimalFace Dog|
> |-|-|-|-|-|-|
> |ADA+DigGAN|36.38|25.42|11.54|35.67|59.98|
> |ADA+NICE|**20.09**|**15.63**|**8.18**|**22.70**|**28.65**|

---

### Official Review · Reviewer_fQTZ · 2023-07-26

**Soundness:** 3 good
**Presentation:** 3 good
**Contribution:** 2 fair
**Rating:** 6
**Confidence:** 3

**Summary:**

The authors present a novel approach called NoIse-modulated Consistency rEgularization (NICE) to solve the challenge of training GANs with limited data. The experiment was conducted on reduced small-scale CIFAR-10, CIFAR-100, ImageNet, and FFHQ datasets. Additionally, they applied their method to low-shot generation tasks, and the results demonstrated state-of-the-art performance.

Overall, the proposed methodology is well motivated and supported by theoretical analysis. The flow of the paper and the writing are easy to follow.


**Strengths:**

1. The paper is written in a clear and well-structured manner, making it easy for readers to follow the presented ideas.

2. Through extensive experiments on various reduced small-scale datasets, the method consistently achieves state-of-the-art performance, demonstrating its effectiveness.

3. The paper is well-grounded in theoretical motivation and effectively validates the proposed methodology through empirical analysis.


**Weaknesses:**

Even though the experimental results support their theory, the ablation study and the experiment for gradient analysis are only based on some specific datasets. Evaluating the theory on more datasets would provide more convincing evidence.

**Questions:**

1.	In Table 1, Table 2, Table 3, and Table 4, what "MA" refers to?

2.	Figure 2 was plotted based on a specific dataset, but the paper does not clearly state which dataset was used to generate this figure. It would be helpful if the authors provide more information about the dataset used for Figure 2.

3.	The notation used in the paper is inconsistent, for instance, with the symbols $\boldsymbol{x}_{\text {real }}$ in page 5 row 205 and $\boldsymbol{x}_r$ in page 5 row 182. Additionally, there is no explanation for the symbols $\boldsymbol{x}_r$ and $\boldsymbol{x}_f$. It would be beneficial for the authors to clarify and provide consistent notation throughout the paper to avoid confusion.

4.	In Equation 8, it is evident that training with features modulated by Gaussian noise leads to regularization of the internal weight by the (2,1) norm. However, it is not clear why the authors did not directly penalize the (2,1)-norm to reduce the Rademacher complexity. It would be valuable if the paper includes a comparison between the Discriminator with adaptive noise (AN) and directly penalizing the (2,1)-norm with adaptive strength $\beta$.

5.	In Equation 9, the training of the discriminator involves minimizing the second-order gradient at fake images. However, Figure 2 (b) shows that the gradient norm at the layer before the classification head for fake images increases. Furthermore, the ablation study with $\mathrm{NICE}{D_f}$ demonstrates improved performance, as well as with $\mathrm{NICE}{G_f}$, but there is no analysis of the gradient norm for the generator.


**Limitations:**

The limitations could be further clarified.

---

> ### Author Rebuttal · Authors · 2023-08-05
>
> # Response to Rev. 1 (fQTZ)
>
> ***We thank the reviewer** for the constructive review and valuable questions that have helped us improve our work.*
> ## 1. What does MA refer to in Tables 1-4?
> We apologize. MA denotes `massive augmentations' (including DA and ADA) first used in:
> > *Generative co-training for generative adversarial networks*, AAAI'22, Cui et al.
>
> DA and ADA are:
> > *Differentiable augmentation for data-efficient GAN training*, NeurIPS'20, Zhao et al.
> \
> *Training generative adversarial networks with limited data*, NeurIPS'20, Karras et al.
>
> ## 2. Which dataset was used in Fig. 2?
> Thank you. This is a 10% data, CIFAR-10, OmniGAN ($d'=256$).
>
> ## 3. Improve notations. What are ${\bf x}_r$ and ${\bf x}_f$?
> Thank you. We have now unified notations as per request. ${\bf x}_r$ and ${\bf x}_f$ are the real and fake samples passed to the discriminator. Indeed, ${\bf x}\_{real}$ is a redundant symbol as it means ${\bf x}_r$.
>
> ## 4. Why the authors did not directly penalize the (2,1)-norm?
>
> * For **the multiplicative modualtion** with the noise, we have:
> \
> $\hat{L}\_{mul\\_noise}:=\hat{\mathbb{E}}\_i\mathbb{E}\_{{\bf z}\sim\mathcal{N}({\bf 1}, \beta^2\mathbf{I}) }||{\bf y}_i-{\bf W}_2({\bf z}\cdot{\bf a}_i)||^2_2=\hat{\mathbb{E}}\_i||{\bf y}_i-{\bf W}_2({\bf z}\cdot{\bf a}_i)||^2_2+\beta^2(\hat{\mathbb{E}}\_i||{\bf a}_i||^2_2)||{\bf W}_2||^2\_{2,1}$,
> where ${\bf z}\cdot{\bf a}_i$ is element-wise multiplication of $\bf z$ and ${\bf a}_i$. This implies that while direct penalty of network weights, $||{\bf W}_2||^2\_{2,1}$, is possible, our approach has a **dynamic penalty** $\beta^2(\hat{\mathbb{E}}\_i||{\bf a}_i||^2_2)$ where variance $\beta^2$ is adapted based on the discriminator decisions but also $\hat{\mathbb{E}}\_i||{\bf a}_i||^2_2$ depends on norms of feature vectors.
> \
> \
> **The importance of such a multiplicative modulation is that semantic contents is not added by noise ${\bf z}$ to the modulated feature vector ${\bf z}\cdot{\bf a}_i$**, i.e., only `active' channels ${\bf a}_i\neq 0$ are modulated: they can be suppressed or magnified according to variance $\beta^2$.
>
> * In contrast, the **additive noise modulation** results in the standard penalty $\beta^2||{\bf W}_2||^2\_{2,1}$ which we suspect the reviewer asks about:
>
>   $\hat{L}\_{add\\_noise}:=\hat{\mathbb{E}}\_i\mathbb{E}\_{{\bf z}\sim\mathcal{N}({\bf 0}, \beta^2\mathbf{I}) }||{\bf y}_i-{\bf W}_2({\bf z}+{\bf a}_i)||^2_2=\hat{\mathbb{E}}\_i||{\bf y}_i-{\bf W}_2{\bf a}_i||^2_2+\beta^2||{\bf W}_2||^2\_{2,1}$,
>   \
>   \
>   While obvious, $\beta^2||{\bf W}_2||^2\_{2,1}$ does not have the **dynamic penalty** term related to feature norms. This also means (based on the derivation) that **the additive noise may change feature semantics**, i.e., ${\bf z}+{\bf a}_i$ may ``activate'' channels which are non-active in ${\bf a}_i$ (features ${\bf a}_i= 0$).
>   \
>   Therefore, **our multiplicative modulator not only controls the Rademacher Complexity (RC), but controls it in a meaningful manner for discriminator**. Only feature semantics that are present in feature vector (that describe object) are modulated while helping control RC. Additive noise would likely introduce semantics that are not present for a given image/object.
>
> * While directly regularizing the network using $\beta^2||{\bf W}_2||^2\_{2,1}$ does not introduce additive noise, it's regularization effect shares similarity with additive noise incorporation. Thus, it's connection to the additive noise suggests that the direct weight regularization has an effect of changing the feature semantics, which will negatively impact the classification accuracy.
>
> * Below we provide **experimental comparisons**. Please note the consistency loss and dual branch from Fig. 1 (right) (main paper) are not used here as that would result in additional penalties on gradient norms:
>
> 10\% CIFAR-10/100 on OmniGAN ($d'=256$), AAN (adaptive additive noise), AWR (adaptive weight regularization), AN (our adaptive multiplicative noise),
> |Method|||10\% CIFAR-10|||10\% CIFAR-100||
> |-|-|-:|:-:|:-|-:|:-:|:-|
> | |Eq.| IS|tFID|vFID|IS|tFID|vFID|
> |OmniGAN| |8.49|22.24|26.33|8.19|45.41|50.33|
> |+AAN | ${\bf a}_i\:={\bf a}_i\\!+\\!{\bf z}; {\bf z}\\!\sim\\!\mathcal{N}(0,\beta^2{\bf I})$ |8.52|20.12|24.65|9.64|37.68|42.01|
> |+AWR | $\beta^2\|\|{\bf W}\|\|^2\_{2,1}$ |8.44|18.42|22.56|9.80|32.05|36.53|
> |+AN| ${\bf a}_i\:={\bf a}_i\\!\cdot\\!{\bf z}; {\bf z}\\!\sim\\!\\mathcal{N}({\bf 1},\beta^2{\bf I})$ |**9.16**|**10.14**|**13.80**|**11.22**|**23.76**|**28.34**|
>
> In addition, **Figure 3 in the rebuttal pdf** shows that AN achives higher classification accuracy than AAN and AWR, demonstrating that multiplicative noise preserves the feature semantics better than the additive noise and the weight regularization.
>
> ## 5. Fig. 2(b) shows the gradient norm at the layer before the classification head for fake images increases.
> This is because when we train the generator, we optimize the generator by minimizing:
> $-h\_\text{AN}({\bf a}\_f)\approx-h({\bf a}\_f)-\frac{\beta^2}{2}\frac{\partial^2 h}{\partial f^2}|\_{{\bf a}_f}f''({\bf a}_f){\bf a}_f^2$
>
> This means the generator is trained to generate images to the point where the gradient norms of discriminator can become large.  Please also see **Resp. 3 Rev. 3 (LrCf)** for discussing of NICE$\_{D_f}$ and NICE$\_{G_f}$ and the **Figure 2 in the rebuttal pdf** for the gradient analysis.
>
> ## 6. Results of different ablation models on CIFAR-100 on OmniGAN ($d'=256$)
>
> Kindly see below reprint of Figure 4(a) which is a table in the main paper.
>
> |Method||10\% CIFAR-100||
> |-|-:|:-:|:-|
> ||IS|tFID|vFID|
> |OmniGAN|8.19|45.41|50.33|
> |+AWD|9.64|37.68|42.01|
> |+AN+AGP|10.72|27.73|32.15|
> |+NICE$_{add}$|9.34|28.59|33.02|
> |+DA|10.16|24.50|28.96|
> |+ADA|11.23|23.11|27.58|
> |+AACR|11.37|21.42|25.76|
> |+NICE|**11.50**|**16.91**|**21.56**|
>
>  Additionally, we provide additional gradient analysis in **Figure 1\&2 in the rebuttal pdf**.

---

### Author Rebuttal · Authors · 2023-08-09

# General Rebuttal

We thank the reviewers' for constructive suggestions and in depth analysis helping us refine our work. We are humbled by such a positive response, and we truly appreciate it.
\
\
**Also, kindly refer to the rebuttal PDF** (**at the bottom of this panel**) for additional figures and tables. Individual rebuttals refer to them in more detail.

# Below we clarify remaining issues for Rev. 3 (LrCf)

## 1. Report the experimental results using only Eq. 12 and Eq. 14 (Rev. 3 (LrCf)).

Below are the results as per request. As expected, stablizing GAN training is the best when all steps of GAN strive to regularize the gradient (Eq.12-14):

|Regularization Eq.||10\% CIFAR-10|||10\% CIFAR-100||Obama|
|-|-:|:-:|:-|-:|:-:|:-|:-:|
||IS|tFID|vFID|IS|tFID|vFID|FID|
|Eq. 12 + 14|9.16|8.69|12.59|11.19|18.80|24.13|29.95|
|Eq. 12 + 13 + 14|**9.26**|**7.23**|**11.08**|**11.50**|**16.91**|**21.56**|**24.56**|

**Figure 2 in the rebuttal PDF** also shows gradient norms of discriminator.

## 2. Does Eq. 1 encompass all GAN objective functions, i.e., the original min-max and non-saturating GANs?  (Rev. 3 (LrCf)).

Certainly, Eq. 1 forms the cornerstone for analyzing the generalization of diverse GAN objective functions. Eq. 1 was introduced by Zhang et al. in "On the Discrimination-Generalization Tradeoff in GANs", ICLR'18.

Their study adeptly showcased that Eq. 1 encompasses a wide range of GANs, including the well-known $f$-GANs. These encompass GANs that minimize the $f$-divergence, which notably encompasses the original GAN formulation.

Additionally, insights from  "Non-saturating GAN training as divergence minimization" (arXiv:2010.08029, Shannon et al.) further bolster this applicability. This work discusses how non-saturating GANs can be approximated via minimizing a specific $f$-divergence, aligning with Eq. 1.

These collective contributions highlight that Eq. 1 provides a unifying lens at GANs, encompassing a wide range of models, including the classic min-max and the non-saturating variants.

---

### Decision · Program_Chairs · 2023-09-21

**Decision:**

Accept (poster)

**Comment:**

1x SA, 1x A, 2x WA, and 1x BA. This paper proposes a noise modulation and regularization scheme for GANs that reduces discriminator overfitting and improves training stability for low data. The reviewers agree on accepting the paper due to its (1) clear presentation, (2) SOTA performance, and (3) thorough theoretical analysis. The rebuttal has addressed their concerns.